# On the choice of finite element for applications in geodynamics. Part II: A comparison of simplex and hypercube elements

Cedric Thieulot[1] and Wolfgang Bangerth[2]

[1]Department of Earth Sciences, Utrecht University, Utrecht, The Netherlands
[2]Department of Mathematics, Department of Geosciences, Colorado State University, Fort Collins, CO, USA

**Correspondence:** C. Thieulot (c.thieulot@uu.nl)

**Abstract.** Many geodynamical models are formulated in terms of the Stokes equations that are then coupled to other equations. For the numerical solution of the Stokes equations, geodynamics codes over the past decades have used essentially every finite element that has ever been proposed for the solution of this equation, on both triangular/tetrahedral ("simplex") and quadrilaterals/hexahedral ("hypercube") meshes. However, in many and perhaps most cases, the specific choice of element does not seem to have been the result of careful benchmarking efforts, but based on implementation efficiency or the implementers' background.

In a first part of this paper (Thieulot and Bangerth, 2022), we have provided a comprehensive comparison of the accuracy and efficiency of the most widely used hypercube elements for the Stokes equations. We have done so using a number of benchmarks that illustrate "typical" geodynamic situations, specifically taking into account spatially variable viscosities. Our findings there showed that only Taylor-Hood-type elements with either continuous ($Q_2 \times Q_1$) or discontinuous ($Q_2 \times P_{-1}$) pressure are able to adequately and efficiently approximate the solution of the Stokes equations.

In this, the second part of this work, we extend the comparison to simplex meshes. In particular, we compare triangular Taylor-Hood elements against the MINI element and one often referred to as the 'Crouzeix-Raviart' element. We compare these choices against the accuracy obtained on hypercube Taylor-Hood elements with approximately the same computational cost. Our results show that, like on hypercubes, the Taylor-Hood element is substantially more accurate and efficient than the other choices. Our results also indicate that hypercube meshes yield slightly more accurate results than simplex meshes, but that the difference is relatively small and likely unimportant given that hypercube meshes often lead to slightly denser (and consequently more expensive) matrices.

## 1 Introduction

Over the past decades, a large number of geodynamics simulation codes have been built on the finite element method. In the specific context of mantle convection and long-term dynamics simulators, the key component of many models that needs to be solved are the Stokes equations for which finite element methods are well suited, but that leaves many choices still to be made: (1) Should the element choice be of Taylor-Hood-type (where the polynomial degree used for the velocity is chosen one higher

than that for the pressure), or a stabilised equal-order element combination, or any of the non-conforming elements? (2) Should the reference cell for the underlying mesh be simplices (triangles or tetrahedra) or hypercubes (quadrilaterals or hexahedra)?

In a first part of this work, see Thieulot and Bangerth (2022), we have extensively compared different hypercube choices for the finite element combination used to discretise the Stokes equations in the context of models that are relevant to geodynamics applications. Our conclusions there were that the lowest-order Taylor-Hood-type element (denoted by $Q_2 \times Q_1$ on quadrilaterals, or $Q_2 \times P_{-1}$ if one uses a discontinuous pressure element that then leads to local mass conservation) are the only ones that produce accurate results in all circumstances. This conclusion is notwithstanding the fact that these elements are not cheap, owing to their higher-order shape functions and the consequent large number of entries in the system matrix. Yet, all other choices we have compared there – specifically, the stabilised $Q_1 \times Q_1$ and the unstable $Q_1 \times P_0$ combinations – were too inaccurate, unstable, and had difficulty representing the hydrostatic pressure component to be competitive. Of course these other choices are widely used in many existing codes, making studies such as Thieulot and Bangerth (2022) useful to inform what the next generation of codes should build on.

At the same time, in Thieulot and Bangerth (2022) we did not investigate whether simplex or hypercube meshes are better suited to the task. Historically, geodynamics has largely settled on the use of quadrilateral or hexahedral ("hypercube") elements – somewhat separate from the rest of the finite element world that has traditionally predominantly used triangular or tetrahedral ("simplex") meshes. The reasons for this deviation are likely rooted in the fact that the geometries of the domains used in geodynamics are largely rather simple: Rectangles and boxes, along with circles, spheres and shells. These geometries present no difficulties to meshing with hypercube cells, whereas the complex geometries frequently used in solid and fluid mechanics can often only reasonably be meshed using mesh generators that create simplex meshes. Still, one could of course also use simplex meshes in geodynamics and in fact many codes have done so over the past decades; see, for example, Barr and Houseman (1996) (BASIL code, $P_2 \times P_1$), Dabrowski et al. (2008) (MILAMIN code, $P_2^+ \times P_{-1}$), Tommasi et al. (2009) (FORGE2005 software, $P_1^+ \times P_1$), Davies et al. (2011) (Fluidity code, $P_2 \times P_1$), Chertova et al. (2014) (SEPRAN, $P_2 \times P_1$), Paczkowski et al. (2014) (COMSOL, $P_2 \times P_1$), de Montserrat et al. (2019) (LaCoDe, $P_2^+ \times P_{-1}$), Jones et al. (2021) (FEniCS project, $P_2 \times P_1$), Ilangovan et al. (2024) (HyTeG framework, $P_2 \times P_1$). It is, therefore, a reasonable question whether that would result in more accurate simulations for the same computational cost, or less costly simulations at the same accuracy.

We are not aware of systematic comparisons between the two choices of reference cell – simplex or hypercube – in the geodynamics literature. Perhaps surprisingly, there is also not a large body of literature on the topic in other disciplines, nor is there a strong oral "lore" in the Scientific Computing community about which of the two approaches is better. In our search for past work, we have found a modestly informative recent publication that clearly illustrates the benefits of quadratic over linear elements, but only a weak preference for triangles/tetrahedra over quadrilaterals/hexahedra (Schneider et al., 2022). That publication also contains references to other, earlier studies in the same direction; it is worth also pointing out Terrel et al. (2012) as another example of studies that compare different elements, though not in great depth and not for applications relevant in geodynamics. On the other hand, while discretisation accuracy matters, so does solver speed. In this regard, modern solver techniques intended to better utilize the power of CPUs over the limitations of memory latency, specifically matrix-free approaches, heavily build on the fact that shape functions on hypercube cells can be written as a tensor product of one-

dimensional functions, and so are naturally more suited to hypercube cells (Kronbichler and Kormann, 2019; Munch et al., 2023).

Regardless of which reference element is better suited, using simplex meshes also opens up a number of other possibilities. Specifically, the number of stable Stokes element combinations for triangles and tetrahedra is substantially larger than it is for quadrilaterals and hexahedra, owing to decades of research on "non-conforming" elements – that is, finite element choices whose basis functions are not continuous, but have a sufficient amount of structural properties (such as being continuous at edge midpoints) that it is not necessary to add specific stabilization terms to the weak formulation of the Stokes equations.[1] Examples of such non-conforming elements include the Brezzi-Douglas-Marini (BDM) element (Brezzi et al., 1985), and the Crouzeix-Raviart nonconforming $P_1$ element,[2] (Braess, 2007). Non-conforming elements have of course also been developed for hypercube meshes – see, e.g., the Rannacher-Turek element (Rannacher and Turek, 1992) or the DSSY element (Douglas Jr et al., 1999). However, in contrast to their relatives defined on simplex meshes, they have not found widespread use and are less often implemented in widely used finite element libraries. Thus, non-conforming elements are generally only considered viable choices on simplex meshes.

In practice, however, non-conforming elements have never found much use in the geodynamics community. As a consequence, while we consider them viable alternatives (and potential targets for future studies) we will not include any non-conforming elements in this work (see also Section 3.3).

We end this overview by mentioning that while we have not found much literature that *quantitatively* compares reference cell and element choices, the book by Gresho and Sani (2000) contains an extensive and excellent overview of the many available choices for the Stokes equations in Sections 3.13.2 to 3.13.6, covering more than 150 pages. Tables 3.13-1 to 3.13-4, along with lengthy comments throughout the section, provide arguments that lead the authors of that book to favor hypercube cells over simplex cells, and to favor Taylor-Hood-type elements over others, based on qualitative arguments and references to the literature from the 1970s, 1980s, and 1990s – opinions that we share, based on the results of this paper and of Thieulot and Bangerth (2022). Yet, the authors also state that the question is not at all trivial, not settled, and in need of systematic quantitative comparisons. In any case, both the book and the literature cited therein exclusively consider the isoviscous Stokes equations which typically has a much smoother solution than the ones we find in geodynamics applications with their highly variable viscosity coefficient. As a consequence, we believe that the comparisons we provide here are useful not only because they are quantitative, but also because they are specific to the kinds of applications we typically encounter in our discipline. We should also mention the paper by Pelletier et al. (1989) that contains qualitative comparisons between Taylor-Hood elements with continuous and discontinuous pressures, and that advocates for the use of discontinuous pressure elements. However, we consider its interpretation difficult in today's context because the meshes used there are quite coarse, and we think it likely that

---

[1]Stability despite lack of continuity for non-conforming elements is in contrast to the "discontinuous Galerkin (dG)" approach in which shape functions are entirely discontinuous and the problem is regularized by introducing penalty terms that ensure that the jumps between cells in the discrete solution are not too large.

[2]The 'nonconforming Crouzeix-Raviart element' must not be confused with the $P_2^+ \times P_{-1}$ element we will discuss below, and that is often also called the 'Crouzeix-Raviart element'.

their recommendations are perhaps no longer as applicable to today's much finer meshes as they were in the historical context nearly 40 years ago when the paper was written.

*Goals of this paper.* Given the setting described above, our goal in this contribution is to compare finite element choices on simplex and hypercube meshes both qualitatively and quantitatively. For hypercube meshes, our previous work in Thieulot and Bangerth (2022) has already indicated that only the Taylor-Hood variants $Q_2 \times Q_1$ or $Q_2 \times P_{-1}$ are reasonable choices, whereas equal-order elements are not. Based on this observation, we then pose the following two questions for the current work:

1. What is the best choice of finite element on simplex meshes?

2. How does the best choice of finite element on simplex meshes compare to the choice of $Q_2 \times Q_1$ or $Q_2 \times P_{-1}$ element on hypercube meshes?

For our numerical comparisons, we will consider both the accuracy and computational cost of a finite element as a function of the mesh size (or number of unknowns) as a criterion. The elements we will consider for simplex meshes include the $P_2 \times P_1$ 'Taylor-Hood'-type element, the $P_2 \times P_0$ (the cheapest stable element with discontinuous pressure), the MINI element $P_1^+ \times P_1$, and the 'Crouzeix-Raviart' element $P_2^+ \times P_{-1}$.

*Outline of the paper.* In the remainder of this paper, we will first briefly state the equations we seek to solve (Section 2). In Section 3 we will then discuss the finite elements one can choose on simplex meshes for the discretisation of the Stokes equations, along with a description of the elements we do and do not compare in this work. Section 4 then provides a numerical comparison of these elements, using a series of benchmarks that illustrate how solutions of geodynamic models often behave. We conclude in Section 5.

## 2 The governing equations

As in the first part of this work, we will here be concerned with the accurate numerical solution of the incompressible Stokes equations:

$$-\nabla \cdot [2\eta\varepsilon(\boldsymbol{u})] + \nabla p = \rho\boldsymbol{g}, \tag{1}$$

$$-\nabla \cdot \boldsymbol{u} = 0, \tag{2}$$

where $\eta$ is the viscosity, $\rho$ the density, $\boldsymbol{g}$ the gravity vector, and we will denote by $\varepsilon(\cdot)$ the symmetric gradient operator defined by $\varepsilon(\boldsymbol{u}) = \frac{1}{2}(\nabla\boldsymbol{u} + \nabla\boldsymbol{u}^T)$. $\Omega \subset \mathbb{R}^d, d = 2$ or $3$ is the domain of interest. Both the viscosity $\eta$ and the density $\rho$ will, in typical applications, be spatially variable; the variability is often introduced through nonlinear dependencies on the strain rate $\varepsilon(\boldsymbol{u})$ and/or the pressure $p$, but the exact reasons are not of relevance to us here: The important point is that these coefficients may vary strongly and on short length scales.

In actual applications, the equations above will be completed by appropriate boundary conditions and will be augmented by additional and often time dependent equations, such as ones that describe the evolution of the temperature field or of the composition of rocks (see, for example, Schubert et al. (2001); Turcotte and Schubert (2012)). This coupling is also not of interest to us here, as is the fact the "true" equations in geodynamics are often compressible – in most cases, the equations above will have to be solved as a "sub-problem" to what one really wants to do, and the efficiency of a discretisation of these equations then translates to a lower bound for the efficiency of solving the outer problem.

## 3 Discretisation

### 3.1 Elements and element combinations

The finite element discretisation of the Stokes equations is complicated by the fact that one cannot choose the piecewise polynomial spaces for velocity and pressure independently. Rather, to obtain a stable discretisation, the pair of spaces needs to satisfy a compatibility condition known as the Ladyzhenskaya-Babuška-Brezzi (LBB) or inf-sup condition (Braess, 2007; John, 2016); the condition, in essence, states that the velocity space must be sufficiently large compared to the pressure space. A common, stable choice is the "Taylor-Hood" space (Taylor and Hood, 1973) that uses piecewise quadratic elements for the velocity, and piecewise linear elements for the pressure.[3]

Yet, there are many more combinations than just the Taylor-Hood choice one could consider (and that are used in practical applications). Specifically, among the conforming velocity elements[4] on simplex meshes, we can consider the following choices:

- $P_1$: The space of piecewise linear, continuous elements;

- $P_1^+$: The same space as above, but with the addition of a "bubble function" on each cell that is a polynomial of order $d+1$ (where $d$ is the number of space dimensions) and is zero on the faces of the cell (see for example Chapter 3.6.1 of John (2016));

- $P_2$: The space of piecewise quadratic, continuous elements;

---

[3]Strictly speaking, Taylor and Hood (1973) did not propose what is today commonly implied by the term "Taylor-Hood" element: They proposed an 8-node serendipity space on quadrilaterals for the velocity components, and the usual 4-node, continuous bilinear space for the pressure. Nonetheless, in today's common usage of the term, a "Taylor-Hood element" is one in which the velocity components are discretised by a piecewise polynomial one degree higher than that used for the pressure, including $Q_2 \times Q_1$ on hypercube cells, but also including $Q_{k+1} \times Q_k$ ($k \geq 1$) on hypercubes, and $P_{k+1} \times P_k$ ($k \geq 1$) on simplices. The term is frequently also used for elements of the same structure, but with a discontinuous pressure space that then guarantees mass conservation. We use the term "Taylor-Hood" herein in this generalized meaning, no longer tied to what the proposed element in Taylor and Hood (1973) may have been. See also (John, 2016, p.98).

[4]The term "conforming" refers to an element choice that respects the continuity properties of the exact solution of a partial differential equation. For example, for solutions of the Stokes equations, the velocity is in the Sobolev space $H^1$ whose elements in 2d are functions that may be discontinuous at individual points, but not along entire lines. Conforming choices for the velocity space must therefore be continuous along faces between cells. At the same time, the pressure solution is only in $L^2$, a space whose members do not need to be continuous, and so any choice of finite element space for the pressure is conforming. Note however that a combination of conforming spaces need not be stable.

|  |  | velocity | | | |
|---|---|---|---|---|---|
|  |  | $P_1$ | $P_1^+$ | $P_2$ | $P_2^+$ |
| pressure | $P_0$ | – | ✓ | ⊘ | ✓ |
|  | $P_1$ | – | ⊘ | ⊘ | ✓ |
|  | $P_{-1}$ | – | – | – | ⊘ |

**Table 1.** A summary of which simplex element combination is stable. A dash indicates that the element combination is not stable, whereas a check mark indicates that it is (John, 2016; Arnold et al., 1984; Reddy and Gartling, 2010). Circles indicate element combinations we consider in this study.

- $P_2^+$: The same space as before, but enriched with cubic bubble functions on each cell.

For the pressure, common choices that match those for the velocity above are:

- $P_0$: The space of piecewise constant and consequently discontinuous elements;

- $P_1$: The space of piecewise linear, continuous elements;

- $P_{-1}$: The space of piecewise linear, but discontinuous elements.

Not all combinations of these are stable (that is, satisfy the LBB condition). Table 1 illustrates which combinations are stable and can consequently be used. At the same time, not all of the combinations are useful; for example, it makes perhaps little sense to use high order polynomials for the velocity when using $P_0$ for the pressure, because the latter might limit the convergence order of the former. As a consequence, we will here only consider a subset of the combinations.

### 3.2 Element combinations used in this study

Concretely, we will show results for the following:

- $P_1^+ \times P_1$: This element is also often called the 'MINI' element. It has not been widely used in the geodynamics community, with the noticeable exception of Zlotnik et al. (2007) in 2d and Tommasi et al. (2009) in 3d.

Because the bubble degrees of freedom only couple to the the other degrees on one cell, it can easily be eliminated from the overall linear system through "static elimination" or "static condensation", making the element as cheap as $P_1 \times P_1$ (but stable!), see for example (Braess, 2007). Since we are mostly interested in questions of accuracy, we will use this element combination but not make use of static elimination in our implementation.

- $P_2 \times P_0$. We could not find any example of this element's use in the (geodynamical) literature. Nonetheless, we decided to include it in this study to document its performance (see previous section) as it is a cheap and stable element with discontinuous pressure, and easily constructable from typical building blocks available in many finite element codes.

- $P_2 \times P_1$: This element is commonly called the 'Taylor-Hood' element (see also footnote 3 above). It is used in geodynamics in, for example, the Fluidity (Davies et al., 2011) and TerraFerma[5] codes (Wilson et al., 2017). It is also used in Schubert and Anderson (1985) and Cuffaro et al. (2020).

  This element corresponds to the widely used $Q_2 \times Q_1$ space on hypercube cells that is used, for example, in the ASPECT code (Kronbichler et al., 2012; Heister et al., 2017).

- $P_2^+ \times P_{-1}$: This element is often referenced as the 'Crouzeix-Raviart' element (Dabrowski et al., 2008; Gresho and Sani, 2000).[6] It has a discontinuous pressure, leading to local mass conservation. The relatively large pressure space requires the augmentation of the $P_2$ velocity space by bubble functions to guarantee stability, but – just like in the case of the $P_1^+ \times P_1$ space above – the bubble degrees of freedom can be removed by static elimination.

  This element is used in geodynamics in, for example, Poliakov and Podlachikov (1992) to study the deformation of the surface above a rising diapir. It is also used in the MILAMIN code (Dabrowski et al., 2008) and in LaCoDe (de Montserrat et al., 2019).

  The closest analog to this element on hypercube elements is $Q_2 \times P_{-1}$ used for example in May et al. (2015).

All of these choices are represented graphically in Fig. 1. In our numerical results below, we will compare these choices against the $Q_2 \times Q_1$ and $Q_2 \times P_{-1}$ elements on hypercube cells, as we have found these to be the best choice in the first part of this study (Thieulot and Bangerth, 2022).

### 3.3 Alternative elements and element combinations, and alternative mappings

There are many more choices one could consider beyond the ones discussed in the previous section. For example, the following come to mind:

- Nearly all of the elements listed above have analogues with higher polynomial degrees. For example, the Taylor-Hood element $P_2 \times P_1$ can be generalized to $P_{k+1} \times P_k$ with $k > 1$; all of these combinations are known to be stable, and at least theoretically result in higher convergence rates. At the same time – see the discussion in Section 3.2 of Thieulot and Bangerth (2022) –, the lack of regularity of solutions in typical geodynamics applications makes these choices unattractive: They are more expensive without delivering higher accuracy because the solution is not smooth enough to actually allow for higher convergence orders. As a consequence, we will not consider higher polynomial degrees herein than the ones mentioned previously.

- There are variations of the spaces above in which a $P_1$ pressure space is enriched by piecewise constants, yielding the $P_1 + P_0$ space. The resulting element, when paired with a sufficiently large velocity space, is then mass conserving.

---

[5]http://terraferma.github.io/

[6]Other authors, for example (Ern and Guermond, 2021, chapter 36), use the term 'Crouzeix-Raviart element' for a different, non-conforming element that is linear but discontinuous, with nodes at edge mid-points. The confusion originates from the fact that Crouzeix and Raviart in the 44 pages of (Crouzeix and Raviart, 1973) introduced a substantial number of elements, including both the one mentioned in the main text and the one of this footnote.

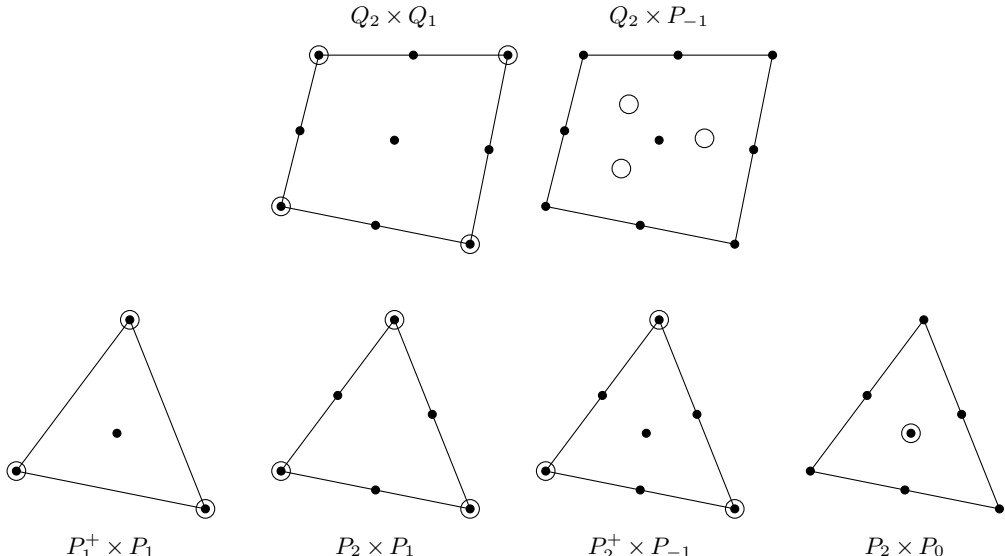

**Figure 1.** A graphical representation of the elements and their degrees of freedom we consider herein. Filled dots indicate locations where velocity degrees are defined, whereas open circles indicate where pressure degrees of freedom are defined. The figure does not reflect whether the shape function associated with a degree of freedom is continuous across cell boundaries.

- Another variation is to replace a $P_2$ velocity space by a $P_1$ space on a once-refined mesh. This is commonly referred to as the "$P_1$iso$P_2$" space (Bercovier and Pironneau, 1979). The original intent in developing this element was to re-use parts of existing implementations, as well as the robustness of linear elements (for example the fact that they always attain their minima and maxima at node points, unlike higher order shape functions).

- There are also numerous non-conforming velocity spaces in which the velocity is not continuous, and that can then either be made convergent through penalty terms, or by requiring structural properties such as that the velocity is at least continuous at face midpoints, see Gresho and Sani (2000) or John (2016) and references therein.

While perhaps useful, these alternatives are not widely used in geodynamics, and we will consequently not consider them herein. Given the conclusions we will come to in Section 5, one can also (retroactively) speculate that at least the non-conforming elements will not be competitive with the best elements we will find in the numerical results in Section 4. This is because most of the nonconforming elements were developed with the specific purpose to scope out how small one can make elements (in terms of degrees of freedom), dating back to a time where that was a prime consideration given how small computer memory was at the time, rather than with the purpose of coming up with accurate and universally robust elements. Indeed, the "small" elements we consider herein will prove to be unacceptable for some reason or other below. Of course, whether the speculation that nonconforming elements are not competitive is in fact true would make for an interesting topic for follow-up work.

A separate issue we defer to a later study is how the choice of mapping from the reference cell (triangles or quadrilaterals in our 2d examples herein) affects the accuracy. For elements with quadratic velocity shape functions, it would not be unreasonable to use quadratic (i.e., "isoparametric") mappings resulting in curved edges. In contrast, herein we only ever use straight-edged elements. Indeed, one could completely separate the polynomial degree of the mapping from that of the finite element – for example, ASPECT by default uses quartic (i.e., "supraparametric") mappings to accurately resolve curved surfaces (Kronbichler et al., 2012). It is not entirely clear how the choice of higher-order mappings would affect accuracy: Practical experience shows that using higher-order mappings results in smaller errors when the geometry is curved (and one might conjecture that this would also be the case if one could resolve internal boundaries). At the same time, there remain open theoretical questions about the stability of the usual Stokes elements when curved boundaries are used, see for example Chilton and Suri (2000). In the end, we believe that the choice of mapping is orthogonal to the choice of element, as we see no reason that an element that is *not* competitive with the best elements we identify here when using straight edges, should *become* competitive when using curved edges. Rather, we consider the current study as a "filter" that allows us to identify which elements are competitive and which are not; we will then leave it to a later study to determine how they can be used with higher order mappings.

### 3.4 Computational setup

For the numerical simulations shown in the following sections, we use the elements mentioned in Section 3.2. Because there is no reason to believe that the elements we choose perform differently in three space dimension, we restrict our computations to two-dimensional benchmarks because (i) these computations are substantially cheaper, (ii) it is far easier to observe convergence rates accurately in 2d: It is possible to reach much higher mesh resolutions and, consequently, get more data points in the asymptotic range where errors strictly follow $\mathcal{O}(h^\alpha)$ rates where $h$ is the mesh size and $\alpha$ describes the convergence rate.

The finite element method requires the computations of integrals, for which we will use quadrature with a number of points that guarantees exact integration as long as coefficients are constant. For example, when using the Taylor-Hood element with quadratic shape functions, we use a quadrature formula with 6 quadrature points per triangle, arranged in the usual fashion of Gauss-type schemes. The result of the finite element integration is then a matrix for the Stokes system that is passed to a linear solver. Although advanced linear solvers are usually preferable for geodynamical codes (e.g. Kronbichler et al. (2012); May et al. (2015); Clevenger et al. (2020); Clevenger and Heister (2021)) we here resort to building the whole Stokes matrix as a sparse array and use a direct solver provided via the SciPy package.[7] None of the computational experiments we perform herein presents the problem of a velocity nullspace; consequently, after solving the linear system, we normalise the pressure by subtracting a constant so that the average pressure is zero. All of these steps were implemented in a Python code written for the purposes of this study.

From the velocity and pressure fields computed via the procedure described above, we can then compute errors by subtracting the exact solutions (where known) and applying appropriate norms. In order to make results comparable to those in Thieulot and Bangerth (2022), we show these errors as a function of the "mesh size" $h$. For a specific cells $K$, we define its size as $h_K = \sqrt{2 \, \text{area}(K)}$ for triangles and $h_K = \sqrt{\text{area}(K)}$ for quadrilaterals. As a consequence, the cell sizes $h_K$ are the same for

---

[7]https://docs.scipy.org/doc/scipy/reference/generated/scipy.sparse.linalg.spsolve.html

a volume subdivided into quadrilaterals and for one in which every quadrilateral is then further subdivided into two triangles; when using corresponding polynomial degrees, these two meshes generally have the same (or approximately the same) number of degrees of freedom, and the definitions of $h_K$ above then guarantee comparability of results. In practice, on structured meshes, all cells have the same $h_K$; on the unstructured meshes we will use, they are approximately equal. We therefore only report results as a function of $h$, which we define as the average value of the $h_K$ values.

Clearly, different elements will have different *costs* for the same value of $h$. However, in practice, the actual cost depends on many specifics of the element choice as well as the linear solver used; consequently, we present our comparisons primarily in terms of the mesh size $h$ rather than the number of unknowns $N$ or any other measure primarily because $h$ is what we used before and because theoretical results about convergence rates readers are likely familiar with are typically shown in terms of powers of $h$. However, Section 4.1 also contains a brief discussion of run times for the various element choices.

In this study we present results obtained on structured and unstructured meshes. Structured meshes are obtained by tessellating the domain with $N_x \times N_y$ quadrilaterals (and in practice setting $N_y = N_x$ for simplicity) as shown in Fig. 2a. For simplex meshes, these quadrilaterals are then cut along a diagonal. In order to avoid very anisotropic meshes and potentially problematic cases where three vertices of a triangle would be on the boundary (see Boffi et al. (2012) or Cioncolini and Boffi (2019) for reasons to avoid this situation), simplex meshes are built so that we vary the direction of splitting quadrilaterals as shown in Fig. 2b.

We create unstructured simplex meshes by creating meshes via the Triangle module which is a Python wrapper around Jonathan Richard Shewchuk's two-dimensional quality mesh generator and Delaunay triangulator library (Shewchuk, 1996, 2014) based on a target mesh size – see Fig. 2c. Sequences of unstructured meshes are always created *de novo*, rather than by refinement of the previous mesh, since successive refinement results in block structured meshes. For the particular case of the SolVi benchmark in Section 4.3, we instruct Triangle to place a number of nodes along a quarter circle to match the discontinuity in the coefficients of the benchmark (see Fig. 2d).

## 4 Numerical results

Having so set the scene, let us now turn to quantitative evaluations of the performance of the elements discussed in the previous section. Specifically, in the current section, we will use carefully selected benchmarks that are both widely used in the literature to assess geodynamics software, and that we have already used (at least in parts) in the first part of this paper. More complete descriptions of the Donea-Huerta, SolCx, and SolVi benchmarks shown in Sections 4.1–4.4, along with visual depictions of their solutions, can be found in the first part of this work (Thieulot and Bangerth, 2022). We refer there for more details, rather than repeating them here. The Rayleigh-Taylor benchmark of Section 4.5 is outlined in its own section. We end with a comparison of convergence rates between the different elements in Section 4.6.

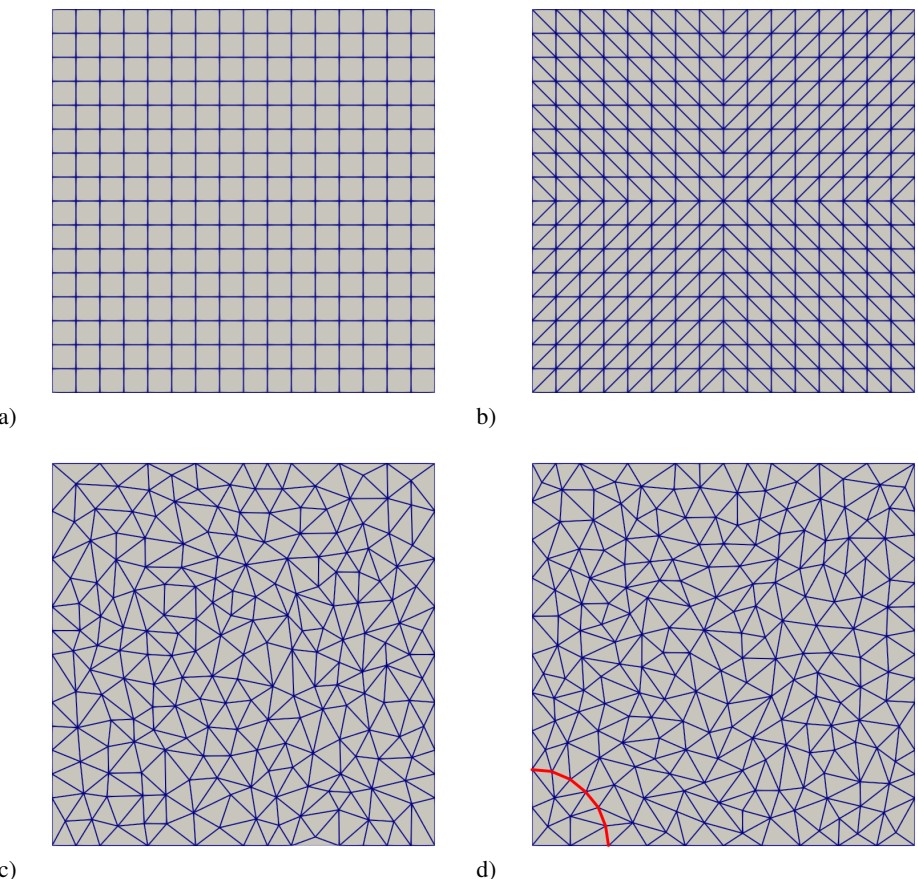

**Figure 2.** Meshes used in this benchmark: a) An example of a structured quadrilateral mesh. b) An example of a structured triangular mesh. c) An example of an unstructured triangular mesh. d) An example of the unstructured triangular mesh used for the SolVi benchmark of Section 4.3; note the nodes (and joining edges) aligned on the quarter circle at the bottom left highlighted in red.

## 4.1 The Donea and Huerta manufactured solution benchmark

The set up for this benchmark – originally described in Donea and Huerta (2003) – considers a situation where the solution is described by smooth polynomials and where the coefficients in the Stokes equations are all constant. The solution is driven by a (nonphysical) gravity field. Given the smooth solution, the different elements ought to all reach their theoretically optimal convergence rate. We use this benchmark, among other reasons, to verify the correctness of our implementations.

We show results in Fig. 3 that illustrate the accuracy with which the various discretisations approximate the exact solution. It shows that – on both structured and unstructured meshes – the discretisations that use piecewise quadratic polynomials for the velocity and linear polynomials for the pressure reach their expected velocity error of $\|\mathbf{u}-\mathbf{u}_h\|_{L_2} = \mathcal{O}(h^3)$, whereas the others only achieve $\|\mathbf{u}-\mathbf{u}_h\|_{L_2} = \mathcal{O}(h^2)$. The latter category includes the $P_2 \times P_0$ discretisation that uses a large number of degrees

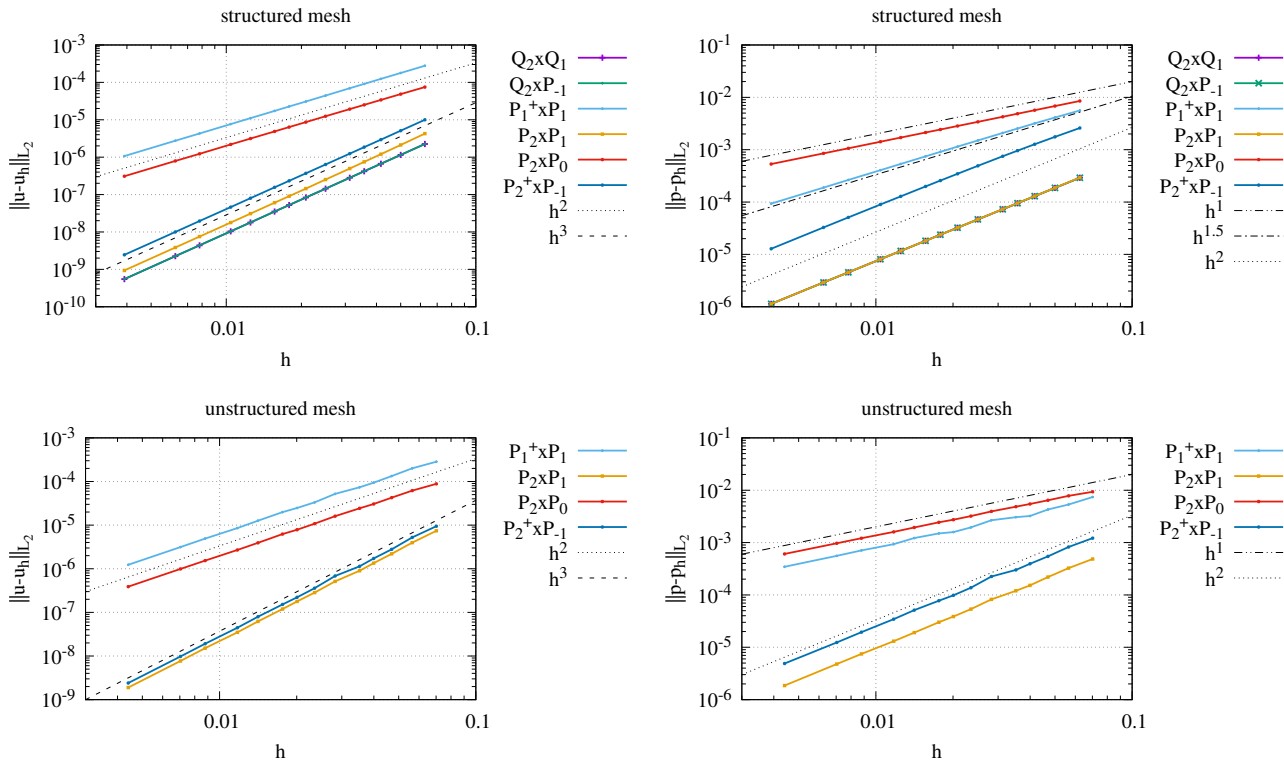

**Figure 3.** Donea and Huerta benchmark: Velocity (left column) and pressure (right column) errors as a function of (average) mesh size for structured (top row) and unstructured meshes (bottom row).

of freedom for the velocity, but achieves an error only smaller than the $P_1^+ \times P_1$ by a factor despite having a a number of degrees of freedom roughly four times higher (in 2d); conversely, it has approximately the same number of degrees of freedom as the $P_2 \times P_1$ elements, but an error about two orders of magnitude larger on the finest meshes. The figure also shows that the $P_2 \times P_0$ element fares even worse in approximating the pressure, being substantially less accurate than the far cheaper $P_1^+ \times P_1$
element (at least on structured meshes).

The figure also shows that at least in terms of accuracy as a function of mesh size $h$ (and consequently number of unknowns), the best performing discretisation is the $P_2 \times P_1$ element, and that it produces errors quite close to the $Q_2 \times Q_1$ and $Q_2 \times P_{-1}$ elements we have found to be best on quadrilaterals. Finally, the figure shows that at least for some of the elements, unstructured meshes can lead to errors nearly an order of magnitude worse than structured meshes with the same mesh size.

In practice, of course, accuracy is only one indicator of performance. A different way to measure how well the different elements perform is to measure the time required to solve a given problem. As a consequence, let us also report run times for the different benchmarks as a function of the number of degrees of freedom, using a laptop with an Intel Core i7-7820HQ CPU run at 2.90GHz, and with 32Gb memory. We caution that unlike the results obtained in the first part of the paper (where we

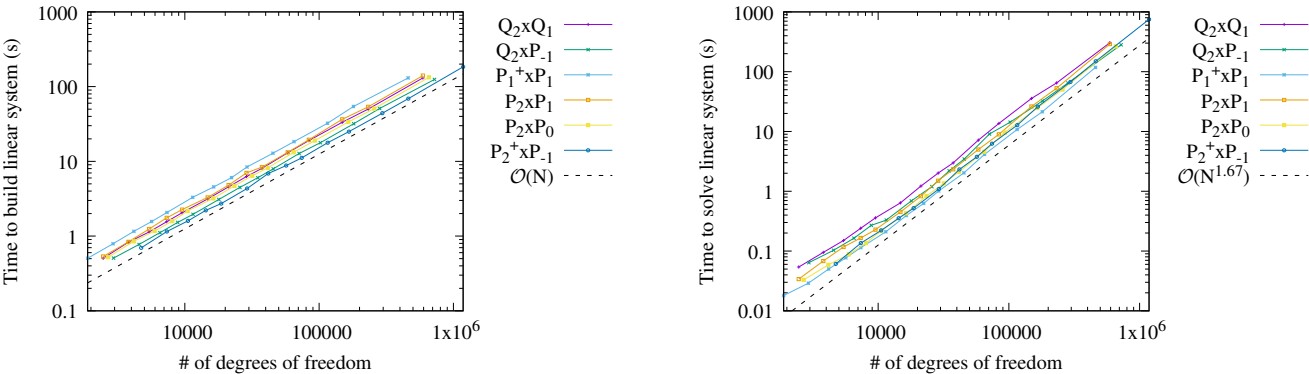

**Figure 4.** Donea and Huerta benchmark: Run time to assemble the linear system (left) and to solve the linear system (right) for the different element combinations, as a function of the number of degrees of freedom on a sequence of meshes.

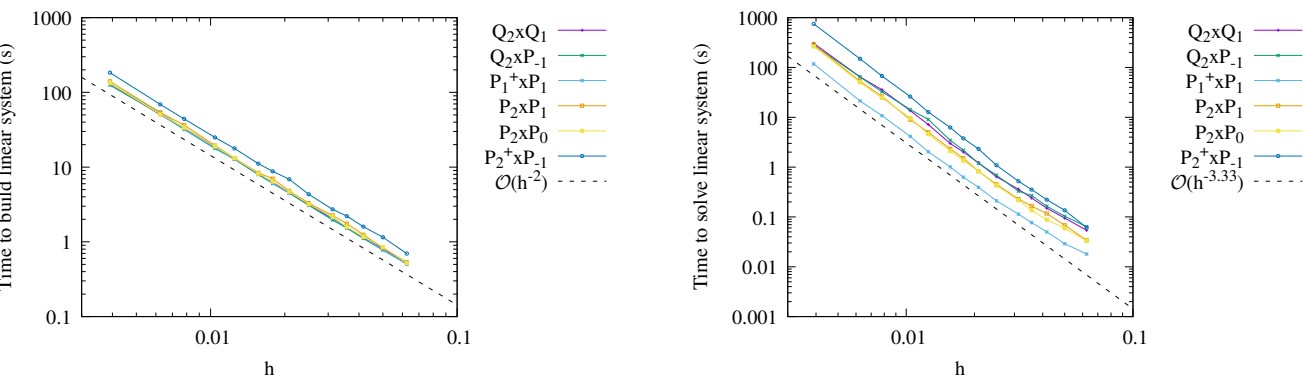

**Figure 5.** Donea and Huerta benchmark: Run time to assemble the linear system (left) and to solve the linear system (right) for the different element combinations, as a function of the mesh size (which is inversely proportional to the square root of the number of cells) on a sequence of meshes.

used the ASPECT code), the run times shown here are obtained with a test code written in Python, rather than a production code written in C++.

Figures 4 and 5 show run times for the two dominant operations – the assembly of the linear system, and the solution of the linear system – as a function of the number of unknowns and the mesh size $h$, respectively. The figures show that the cost of both of these operations is, in essence, a function of the number of degrees of freedom of an element combination, rather than the specific details of how an element's shape functions are defined. As a consequence, the costs of different elements only differ by a (modestly-sized) constant factor, rather than leading to different *rates*. What this also implies is that if we had shown the results of, say, Figure 3 as a function of run time instead of mesh size $h$, curves would only have been moved up and

down, but they would have retained their relative convergence rates and elements shown there with a higher convergence rate will also have a higher convergence rate as a function of run time. A secondary observation is that the run times for the various Taylor-Hood-type elements are not substantially different from each other; as a consequence, run time is not a criterion that will help us choose one of these variants over the others.

Similar observations will hold for the benchmarks in the following sub-sections, and we will consequently omit comparisons of run times there.

## 4.2 SolCx

SolCx is a substantially more difficult benchmark to solve since it involves a viscosity that jumps by a factor of $10^6$ along the vertical mid-line of the domain. This results in a nearly discontinuous pressure as well as a kink in the velocity along this line. With properly aligned meshes, some elements can resolve these singularities, though this of course makes the benchmark not representative of real-world situations where the locations and directions of jumps in the viscosity of geodynamic models can typically not be predicted a priori and may change with time. Elements using continuous pressures consequently exhibit poor convergence. This benchmark is widely used in many geodynamical papers (e.g. Zhong, 1996; Duretz et al., 2011; Kronbichler et al., 2012; Thielmann et al., 2014; de Montserrat et al., 2019; Thieulot and Bangerth, 2022).

Figure 6 shows the approximation errors we obtain for this benchmark. It illustrates the difficulties elements with continuous pressure (such as $Q_2 \times Q_1$ and $P_2 \times P_1$, specifically as opposed to $Q_2 \times P_{-1}$ and $P_2^+ \times P_{-1}$) have with this benchmark: They all only achieve a convergence rate of $\mathcal{O}(h^{0.5})$, reflecting the lack of regularity in the exact pressure; notably, the convergence rate for elements using a continuous piecewise linear pressure is even worse than for the $P_2 \times P_0$ element that uses (discontinuous) piecewise constant pressures.

## 4.3 SolVi

Of course, the difficulties of the SolCx benchmark of the previous section are somewhat artificial, given that the discontinuity in the viscosity is along a vertical line that is easily matched by the mesh (if desired). In other words, while the struggles of elements with continuous pressure are real, the fact that elements with discontinuous pressures work well on such meshes could be considered a lucky break because the jump in viscosity is aligned with the discontinuity of pressures along cell interfaces – at least on structured meshes with an even number of cells per coordinate direction, as we use here.

At the same time, this is perhaps not so. The bottom row of Fig. 6 already suggests that elements with discontinuous pressure spaces can adequately resolve the discontinuous pressure even on unstructured meshes, where the jump in viscosity in the SolCx benchmark is no longer aligned with cell interfaces. The SolVi benchmark we consider in this section illustrates this in more detail. It models a situation where the viscosity inside an inclusion is 1000 times larger than outside the inclusion, and where we make no attempt at resolving this boundary with the structured mesh – similar to realistic situations of slab subduction or other cases of large and perhaps dynamically changing viscosity jumps that cannot practically be resolved using the meshes in use. The setup is identical to the one in Thieulot and Bangerth (2022) although here we only model one quadrant of the problem: The domain is the unit square $(0,1)^2$, the inclusion is centered on the origin, and the analytical velocity is prescribed

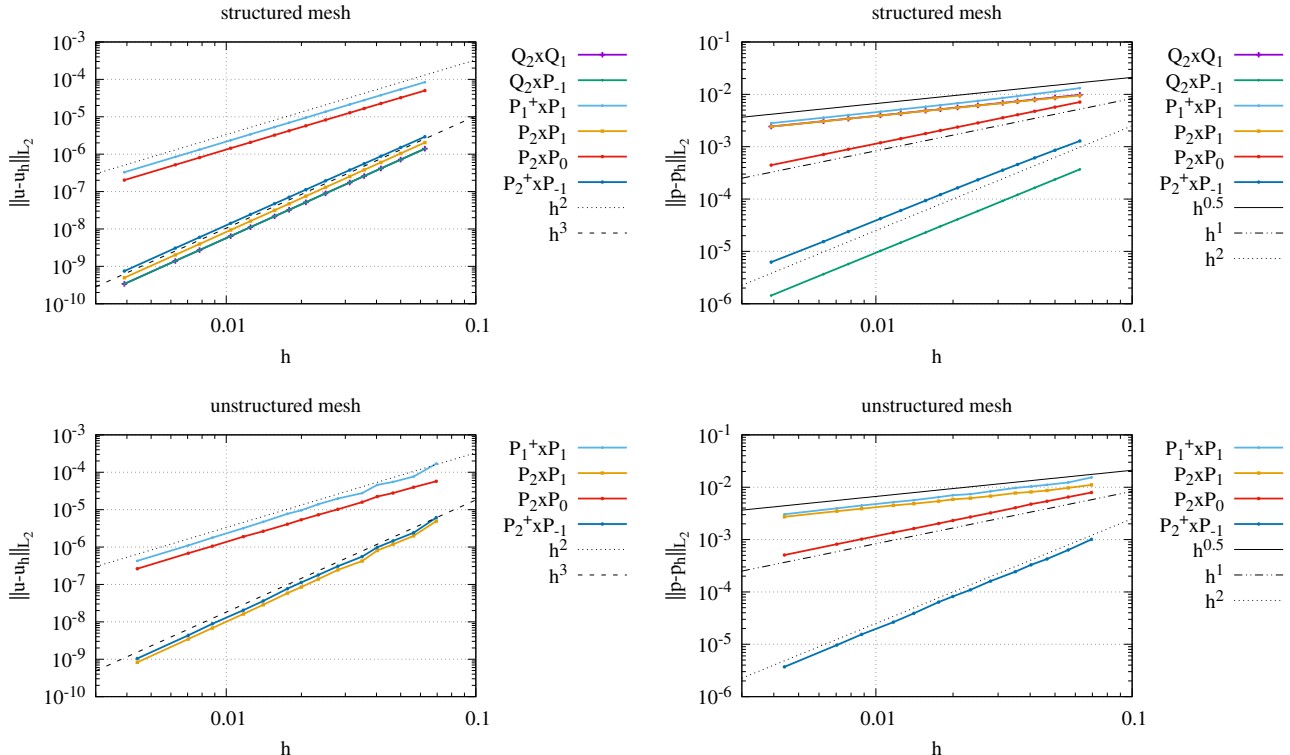

**Figure 6.** SolCx: Velocity (left column) and pressure (right column) errors as a function of (average) mesh size for structured (top row) and unstructured meshes (bottom row).

on all sides. Schmid and Podlachikov (2003) derived a simple analytical solution for the pressure and velocity fields for this case, which was subsequently used in many other publications (Deubelbeiss and Kaus, 2008; Suckale et al., 2010; Duretz et al., 2011; Kronbichler et al., 2012; Gerya et al., 2013; Thielmann et al., 2014; de Montserrat et al., 2019).

We show results for the errors in velocity and pressure in Fig. 7. In the case of structured meshes (the top row of the figure), the figures show that the lack of regularity in the solution, coupled with the fact that the line where this singularity occurs is not captured by the mesh, leads to a situation where all elements only obtain the convergence rate allowed by the solution, rather than based on their polynomial degrees. Indeed, the quality of approximation is largely determined simply by the number of degrees of freedom an element can offer for a given mesh size $h$.

For unstructured meshes, we use the modified procedure shown in Fig. 2d to obtain a triangular mesh whose edges are aligned with the discontinuity of the viscosity – an approach that is admittedly artificial and would not be possible in "real" applications. The corresponding results are shown in the bottom row of Fig. 7. They show that the alignment of cell edges to the discontinuity can recover one order of convergence for the velocity (from $\mathcal{O}(h)$ to $\mathcal{O}(h^2)$ for all of the elements we

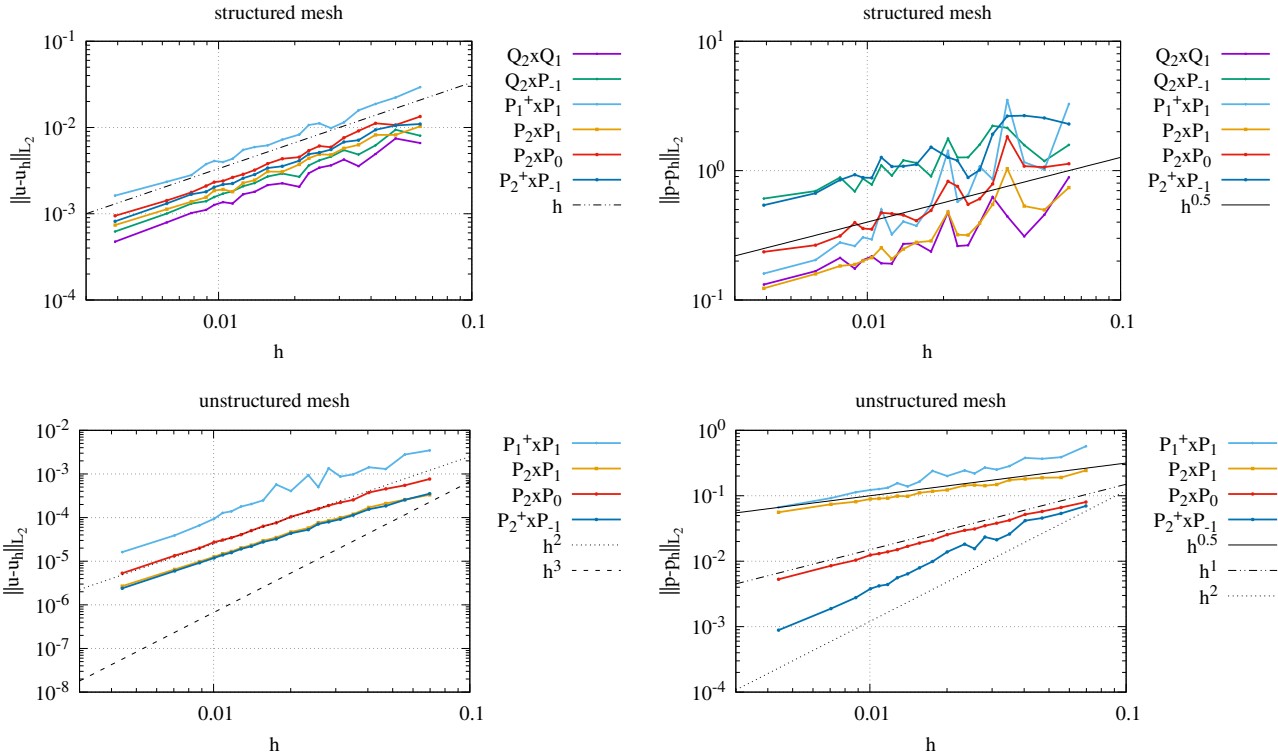

**Figure 7.** SolVi: Velocity (left column) and pressure (right column) $L_2$ errors as a function of (average) mesh size for structured (top row) and unstructured meshes (bottom row).

investigated), and up to one order of convergence for the pressure if one uses discontinuous pressure elements. Yet, even with these aligned meshes, none of the elements achieves its optimal convergence rate.

A comparison of the curves for the structured meshes shows that, for this complex situation, the Taylor-Hood elements $Q_2 \times Q_1$ and $P_2 \times P_1$ fare the best, at least as far as "error for a given mesh size" is concerned. For the very specific discontinuity-aligned unstructured meshes, the $P_2^+ \times P_{-1}$ pair emerges as the overall best with a quadratic pressure error convergence. The observations from these experiments also support the assertion in Section 3.3 (as well as the conclusions of Thieulot and Bangerth (2022)) that higher-order Taylor-Hood elements (i.e., $Q_k \times Q_{k-1}$ or $Q_k \times P_{-(k-1)}$ on hypercubes, and $P_k \times P_{k-1}$ or $P_k^+ \times P_{-k}$ on simplices, in both cases with $k > 2$) would not yield better convergence orders despite their additional cost and are therefore not worth investigating further for geodynamics applications. This justifies why we do not consider them for this study.

## 4.4 The sinking block

In the SolCx and SolVi cases, the difficulty is driven by a discontinuous coefficient (the viscosity) in the differential operator of the Stokes equations (1)–(2). In contrast, for the sinking block benchmark, one considers a situation where a square part of the domain differs not only in viscosity, but also in density from the surrounding material – that is, in the right hand side of the equation. This results in singularities in the solution at the edges of the inclusion that have a qualitatively different behavior than that one observes in the SolCx and SolVi benchmarks. Similar or identical benchmarks can be found, for example, in May and Moresi (2008), Gerya (2019), Thieulot (2011), Mishin et al. (2022), and Schuh-Senlis et al. (2020). The current benchmark also involves having to deal with buoyancy forces (that is, a non-trivial hydrostatic pressure) that are of course the driving force for many effects in geodynamics and whose resolution is therefore important; we have found in the first part of this paper that dealing with buoyancy presented substantial problems to the stabilized $Q_1 \times Q_1$ element.

In the current benchmark, we consider a "sinker" inclusion that has a density $\rho_{\text{sinker}} = \rho_{\text{fluid}} + \delta\rho$ and viscosity $\eta_{\text{sinker}} = \eta^*\eta_{\text{fluid}}$. Boundary conditions are free slip on all sides and gravity is given by $\boldsymbol{g} = -\boldsymbol{e}_y$. The domain is the unit square and we set $\rho_{\text{fluid}} = 1$ and $\eta_{\text{fluid}} = 1$. The sinker is a square of size $0.25 \times 0.25$ centered at $(x_s, y_s) = (0, 0.75)$. As explained in Thieulot and Bangerth (2022), "in a geodynamical context, this setup could be interpreted as a detached slab ($\delta\rho > 0$) or a plume head ($\delta\rho < 0$). As such its viscosity and density can vary (a cold slab has a higher effective viscosity than the surrounding mantle while it is the other way around for a plume head)."

We consider two cases: (1) The fluid and the sinker densities are as described above (the "full density" case); (2) The fluid has zero density and the density of the block is set to $\rho_{\text{sinker}} = \delta\rho$ (the "reduced density" case). The two cases of course lead to the same exact velocity field, but differ in the fact that the pressure field contains a "hydrostatic" (or, in the current context, "lithostatic") component only in the first case, whereas the background fluid (having zero density $\rho_{\text{fluid}}$) does not contribute to the pressure field in the second case. Even though the difference between the two cases is only the addition of a pressure that grows linearly with depth, the discretised equations may show an element-dependent behavior. For example, it is clear that resolving a linear pressure with an element that uses piecewise constant pressures (such as the $P_2 \times P_0$ element) will incur a substantial accuracy penalty; likewise, as shown in Thieulot and Bangerth (2022), stabilized elements yield different solutions based on whether or not the hydrostatic pressure is included.

In order to evaluate the accuracy of different elements for this benchmark, we will make use of the observation shown in Appendix A.2 of Thieulot (2011): While one can independently vary $\eta_{\text{fluid}}, \rho_{\text{sinker}}, \eta_{\text{sinker}}$, and measure $|v_y|$ in the middle of the sinker for each combination, the quantity $v^* = |v_y|\eta_{\text{fluid}}/\delta\rho$ is found to be a function of only the ratio $\eta^* = \eta_{\text{sinker}}/\eta_{\text{fluid}}$. At high enough mesh resolution, all data points then collapse onto a single line (but this may not be the case on coarse meshes: different values of the material constants may correspond in the same $\eta^*$ but numerically result in different values of $v^*$). Similarly, the normalised pressure $p^* = p/\delta\rho g L_b$ measured in the middle of the block is, on sufficient fine meshes, a function of $\eta^*$ only.

We will therefore show figures that report the computed values of $v^*$ and $p^*$ as a function of $\eta^*$, for all six elements. For each $\eta^*$, we show data for $\delta\rho/\rho_{\text{fluid}} \in \{0.25\%, 1\%, 40\%\}$. As mentioned, the values of $v^*$ and $p^*$ obtained with these three density ratios *should* be the same, but are *not* the same on coarse meshes; however, this is only visible in the figures for the $P_2 \times P_0$

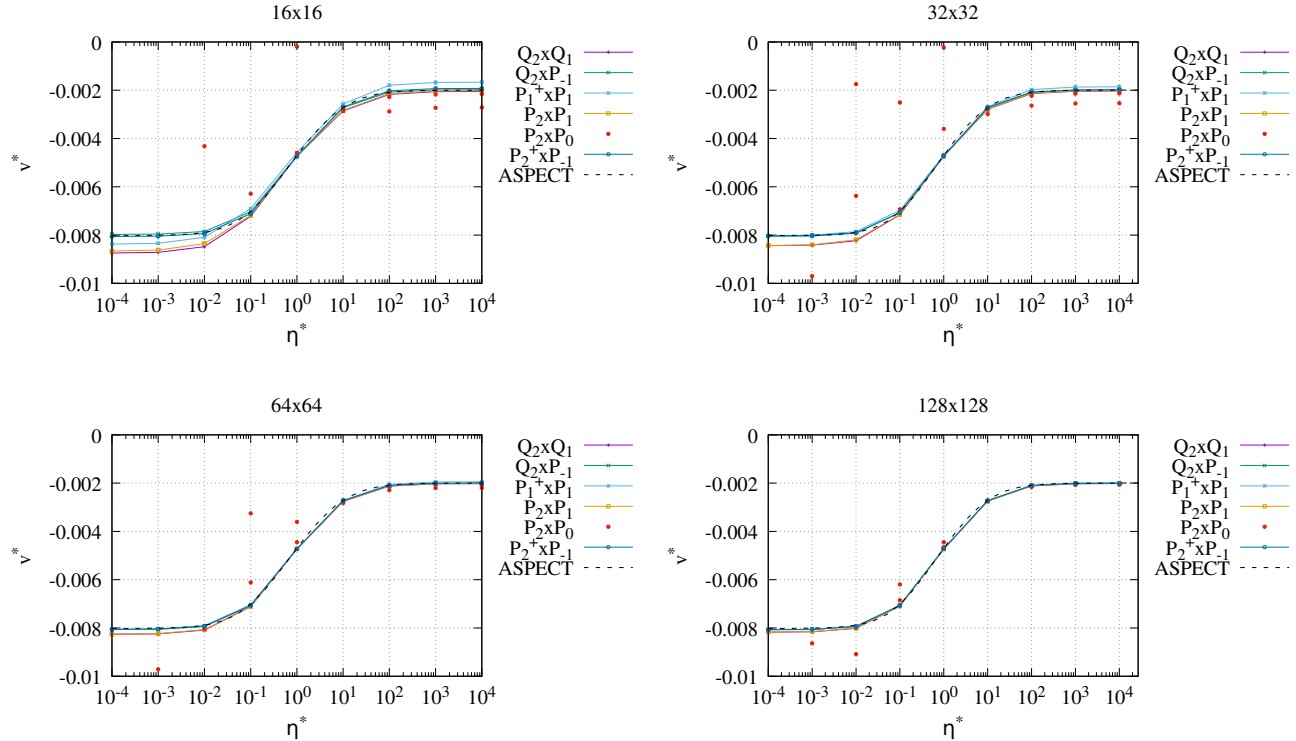

**Figure 8.** The sinking block benchmark with full densities: Normalised velocity $v^*$ in the middle of the block (obtained for three density ratios $\delta\rho/\rho_{\text{fluid}}$) as a function of viscosity ratio $\eta^*$. Each panel corresponds to a different mesh resolution. For the $P_2 \times P_0$ element, some of the data points fall outside of the range of the plots. (See the main text for an explanation of the scattered red dots for the $P_2 \times P_0$ element.) For reference, we also show results obtained with ASPECT on a $256 \times 256$ mesh.

element where for each $\eta^*$ up to three different values of $v^*$ (one for each value of $\delta\rho/\rho$ considered) are apparent. We here restrict ourselves to structured meshes with resolutions $16^2$, $32^2$, $64^2$, and $128^2$, so that element edges align with the boundary of the block.

### 4.4.1 Full density

Fig. 8 shows results for all elements and four different mesh resolutions for the case where we include the lithostatic pressure in the model. We find that, as we increase the mesh resolution, all elements but the $P_2 \times P_0$ converge to reference results obtained with the ASPECT code at $256 \times 256$ with the $Q_2 \times P_{-1}$ element. Because the overall pressure is dominated by the lithostatic component that grows linearly with depth, it is not surprising that the $P_2 \times P_0$ has a hard time approximating the pressure well; the figures show that this also translates to a poor approximation of the normalised velocity $v^*$. This error becomes smaller the

larger $\eta^*$ becomes since $\eta^*$ is a measure of the ratio of the dynamic to the lithostatic pressure.

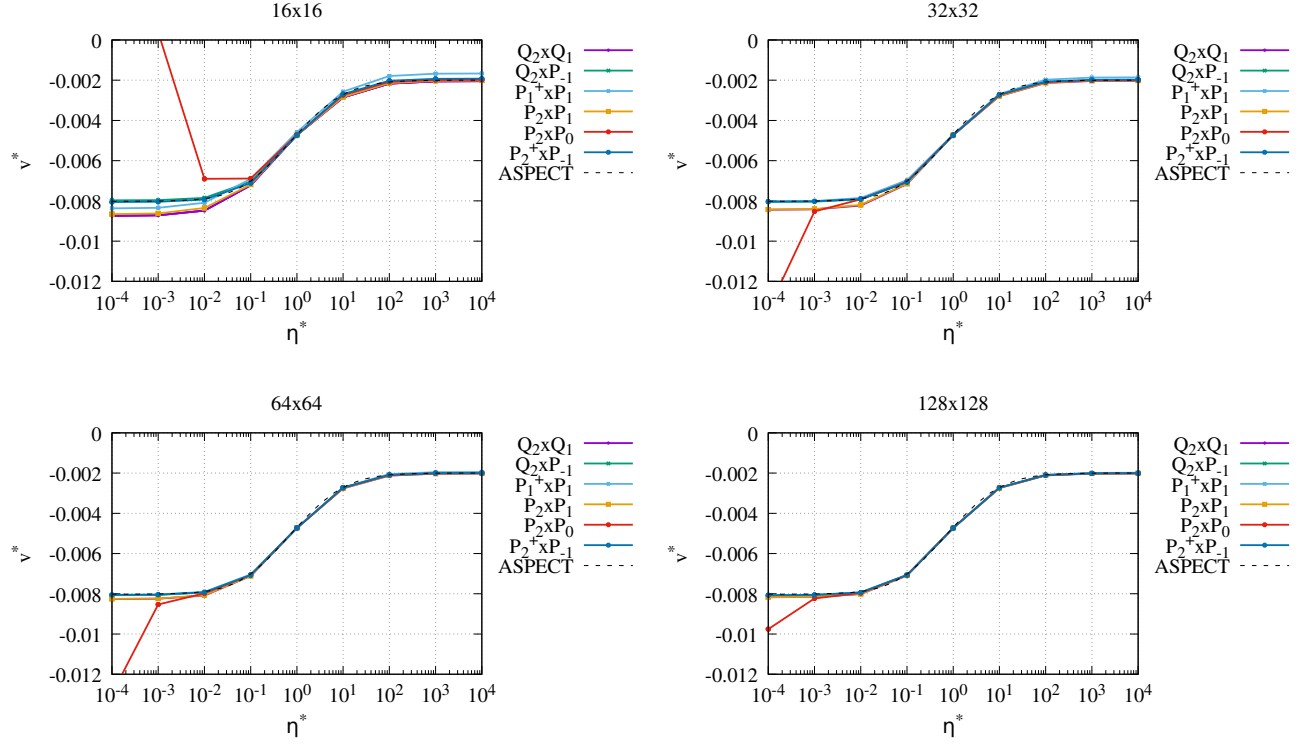

**Figure 9.** The sinking block benchmark with reduced densities: Normalised velocity $v^*$ as a function of viscosity ratio $\eta^*$ for various resolutions.

### 4.4.2 Reduced density

In the second case, where the density outside the inclusion is zero, the lithostatic pressure is absent and we can investigate both the dimensionless velocity (Fig. 9) and pressure (Fig. 10) in the middle of the block.

While the figure shows that the $P_2 \times P_0$ element has recovered some of its accuracy in approximating the velocity, it is
400 unable to provide an accurate approximation of the pressure. A comparison of the convergence behavior (going from coarse to fine meshes) shows that the $P_1^+ \times P_1$ element also behaves pretty poorly. The remaining elements are all of Taylor-Hood type; of these, the $P_2 \times P_1$ element with continuous pressure is substantially more accurate than the $P_2^+ \times P_{-1}$ element with discontinuous pressure.

### 4.5 Rayleigh-Taylor wave benchmark

As a final comparison, we have also carried out the buoyancy-driven Rayleigh-Taylor wave instability benchmark found, for example, in Gerya (2019), Deubelbeiss and Kaus (2008) and Thieulot (2011), and for which an analytical solution of the initial

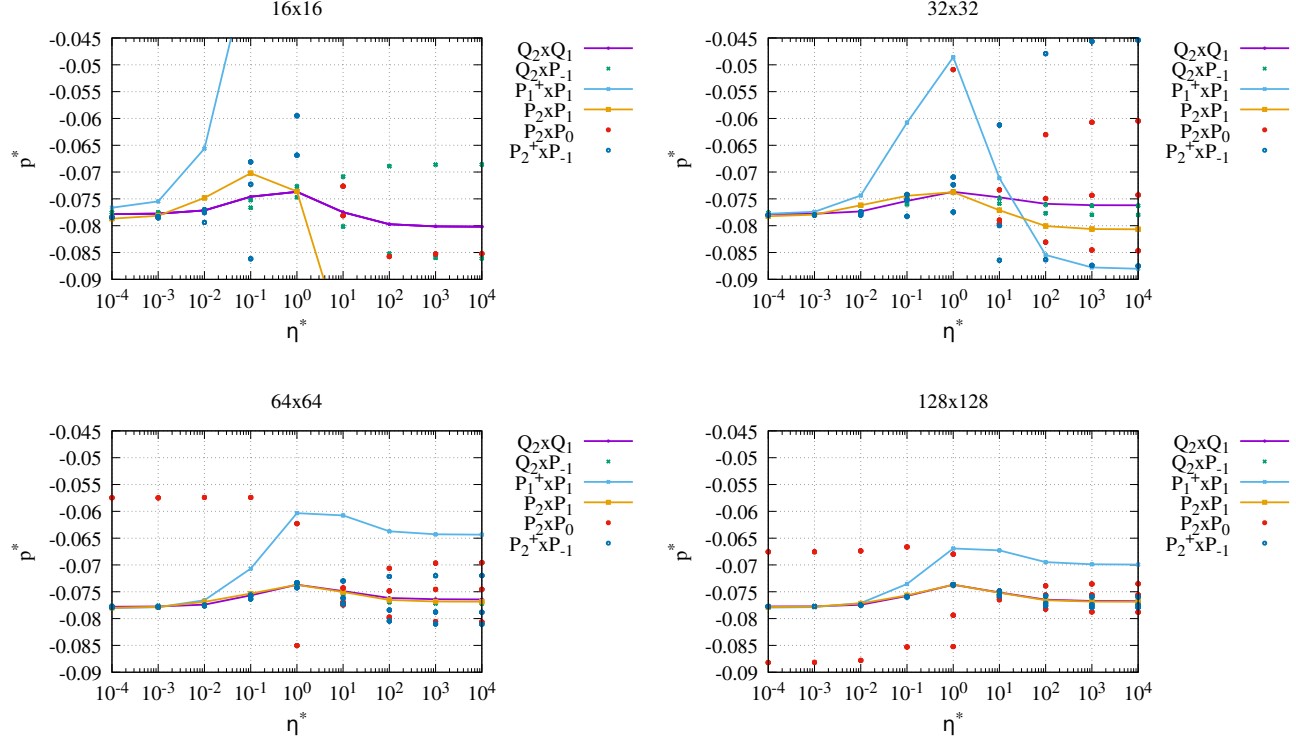

**Figure 10.** The sinking block benchmark with reduced densities: Normalised pressure $p^*$ in the middle of the block as a function of viscosity ratio $\eta^*$ for various resolutions. For the $P_2 \times P_0$, $P_2^+ \times P_{-1}$ and $Q_2 \times P_{-1}$ elements with their discontinuous pressure spaces, we show $p^*$ at several slightly displaced points $(x_s \pm \delta x, y_s \pm \delta y)$. For the $P_2^+ \times P_{-1}$ and $Q_2 \times P_{-1}$ elements the difference is not visible at high resolution, values for the $P_2 \times P_0$ element (red dots) fall outside the range shown here at low resolution and still show substantial differences at high resolution.

growth rate can be found in Ramberg (1968). The benchmark consists of a two-layer system in a box of size $L_x \times L_y$ driven by gravity. A layer of fluid 1 (with viscosity and density $\eta_1, \rho_1$, and thickness $h_1 = L_y/2$) overlies a layer of fluid 2 (with viscosity and density $\eta_2, \rho_2$, and thickness $h_2 = L_y/2$). The interface between the two layes is disturbed by a sinusoidal displacement

characterized by its amplitude $\Delta = 0.01$ and wavelength $\lambda = L_x/2$. No-slip boundary conditions are imposed on the top and the bottom of the domain, while free slip is imposed on the sides. Gravity is set to $\boldsymbol{g} = -\boldsymbol{e}_y$. We use a mesh that is slightly distorted so as to accommodate the sinusoidal interface; however, we use straight element edges in keeping with the other benchmarks solved in this contribution. In our experiments, we specifically choose $L_x = L_y = 1$.

The non-horizontal interface in the set up leads to diapiric growth (illustrated in Fig. 11) whose initial vertical velocity $v$, at

points where it is maximal, can be shown to satisfy the analytic relationship $v = -\Delta K \frac{\rho_1 - \rho_2}{2\eta_2} h_2 g$ with $K$ being a dimensionless

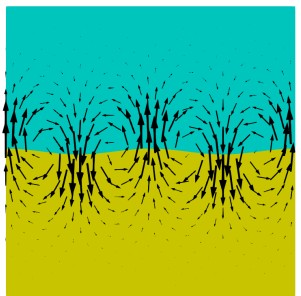

**Figure 11.** Rayleigh-Taylor wave benchmark. The figure shows the two layers and the velocity field that results from their unequal densities, as obtained on a $64 \times 64$ mesh using $Q_2 \times Q_1$ elements, with $\eta_2 = 10^2$.

growth factor that depends on $\phi_1$, $\phi_2$, $\eta_1$ and $\eta_2$ (see Gerya (2019)). Instead of targeting a specific node in the domain, we evaluate $v$ by taking the maximum vertical velocity $|v_y|$ in the domain.

We then solve this benchmark with $\eta_1 = 1$, $\rho_1 = 1.1$, $\rho_2 = 1$ and we vary $\eta_2$ between $10^{-2}$ and $10^2$ and computationally determine the vertical growth velocity at the initial time for all six element pairs and for various resolutions.

We show results in Fig. 12. We find that all elements but the $P_2 \times P_0$ perform as expected for the range of explored viscosity values $\eta_2$: the obtained velocities fall on the analytical dashed line, with fairly little variation between element combinations that is only visible on the coarsest mesh. On the other hand, the results obtained with the $P_2 \times P_0$ combination are far from the exact values; we find that with increasing resolution, obtained velocities get closer to the analytical values (especially for $\eta_2 > 1$), but even at a resolution of $256 \times 256$ elements, $v$ is more than a factor 2 off for $\eta_2 = 0.01$. This behavior is consistent with what we have found for the sinking block benchmark in Section 4.4.

In order to elucidate the underlying reasons, we have re-run this experiment with reduced densities (see Section 4.4) where we choose $\rho_1 = 0.1$ and $\rho_2 = 0$; in other words, we have subtracted a constant from the densities in both layers since flow is only driven by density *differences*, not densities themselves. This results in a different pressure field but the same velocity. We find that in this case the obtained velocities for the $P_2 \times P_0$ converge much faster to the analytical ones over the entire range of viscosities, as shown in Fig. 13. This is again in line with the observations for the sinking block benchmark, and also matches our findings in Thieulot and Bangerth (2022) that constant-pressure elements perform poorly in buoyancy-driven flow experiments where the lithostatic pressure is dominant.

### 4.6 A quantitative comparison of convergence rates

For three of the benchmarks shown in the previous sub-sections, an analytic solution is available that allowed us to compute errors. For these cases, we can also compute error rates in the $L_2$ norm, namely i.e. $\|u - u_h\|_{L_2} \propto h^\alpha$ and $\|p - p_h\|_{L_2} \propto h^\beta$. Generally, for Taylor-Hood-type elements with polynomial degree $k$ for the velocity, one would expect $\alpha = k + 1$ and $\beta = k$

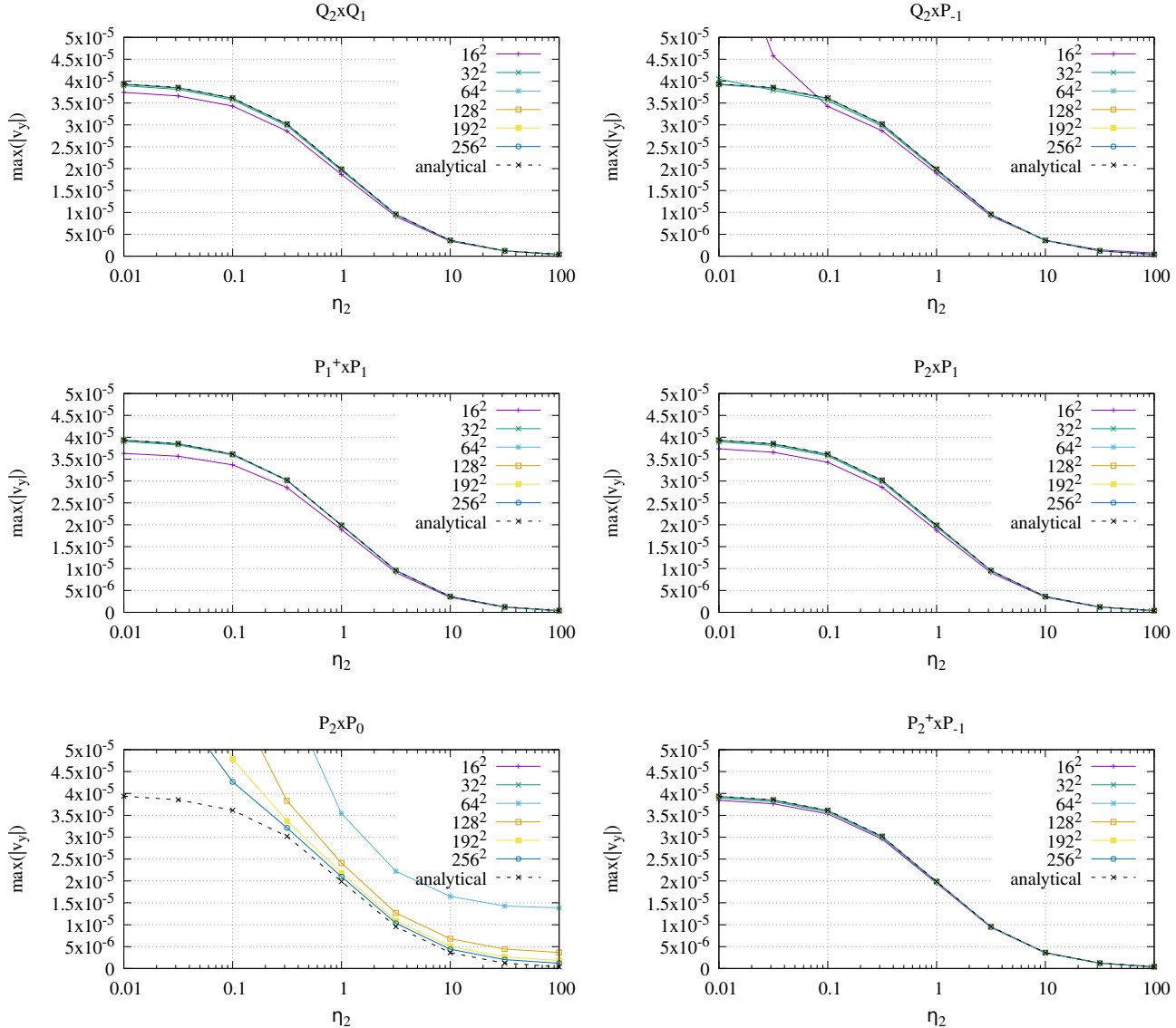

**Figure 12.** Rayleigh-Taylor wave benchmark. For each of the six element combinations considered in this paper, we show the maximum (absolute) vertical velocity $|v_y|$ as a function of viscosity $\eta_2$ for several different mesh resolutions.

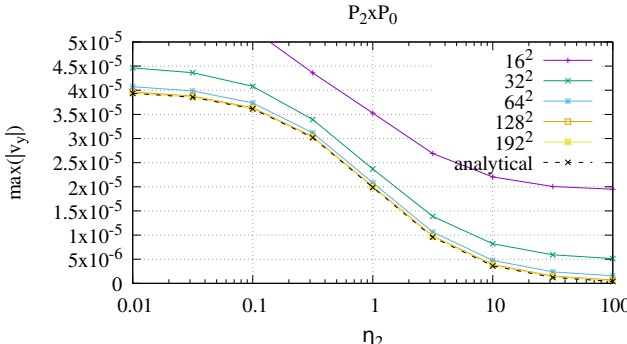

**Figure 13.** Rayleigh-Taylor wave benchmark. We show the maximum (absolute) vertical velocity $|v_y|$ as a function of viscosity $\eta_2$ for several different mesh resolutions for the $P_2 \times P_0$ element. Compared with the bottom left panel in Fig. 12, we here use *reduced* densities, as explained in the main text.

if the solution is smooth, but not all elements always achieve this rate and the rate is also limited by the smoothness of the solution – see the discussion in Sections 3.1 and 3.2 of Thieulot and Bangerth (2022).

We summarize the rates we observe in our computations in Table 2, along with the optimal rate one would expect theoreti-
440 cally for each of these elements. The table illustrates that in cases where the solution is smooth, the Taylor-Hood-type elements achieve a higher order of convergence and, consequently, will be asymptotically more efficient than the other elements. (In practice, the results of the previous sections as well as the first part of this paper show that the Taylor-Hood-type elements are already more efficient for rather coarse meshes.) This observation will apply to the large parts of the domain in geodynamics simulations where the viscosity varies smoothly. The second observation one can draw from the table is that for cases where
the solution is not smooth because the viscosity or density is discontinuous, *all* discretizations take a hit (unless the mesh is aligned with the discontinuity) and convergence rates are limited by the regularity of the solution.

We end this section by noting that we also computed solutions to the SolKz benchmark (Zhong, 1996) that, like the Donea-Huerta benchmark, has a smooth solution and that has been widely used in the community for similar purposes (Duretz et al., 2011; Kronbichler et al., 2012; Gerya et al., 2013; de Montserrat et al., 2019). The results are very similar to those of the Donea-
450 Huerta case. We have also run the benchmark described in John (2016, p. 752), with results matching those provided there. In both cases, the results confirm the correctness of our implementation but do not provide any insight not already available from the benchmarks shown above; we have consequently chosen not to show these results in this contribution.

Finally, in the first part of this paper, we followed our results on benchmarks with a more concrete geodynamic application. The observations there reinforced the conclusions we had drawn based on benchmarks. Based on the results of the current
paper, we see no reason to believe that solving the same application again with triangular meshes would result in any different outcomes than reported there, and we consequently omit it here.

**Table 2.** Observed convergence rates for the three benchmarks for which an analytic solution is available, along with the theoretically predicted optimal convergence rate for each of the elements assuming sufficiently smooth solution. Each entry in the table consists of a pair $\alpha/\beta$ of convergence rates for the $L_2$ norms of the error in the velocity/pressure, respectively. "struct.": structured meshes; "unstruct.": unstructured meshes (for simplex elements only). Note that the optimal pressure convergence rate for the MINI element $P_1^+ \times P_1$ depends on the type of mesh; on general meshes, standard finite element theorey predicts it to be 1, but in certain conditions can be up to 1.5 as observed for the Donea-Huerta benchmark (see Cioncolini and Boffi (2019) and John (2016, p.157) for experimental evidence, and Eichel et al. (2011) for an earlier theoretical investigation).

| | | Optimal | Donea-Huerta | | SolCx | | SolVi | |
|---|---|---|---|---|---|---|---|---|
| | | | struct. | unstruct. | struct. | unstruct. | struct. | unstruct. |
| hypercube | $Q_2 \times Q_1$ | 3/2 | 3/2 | – | 3/0.5 | – | 1/0.5 | – |
| elements | $Q_2 \times P_{-1}$ | 3/2 | 3/2 | – | 3/2 | – | 1/0.5 | – |
| simplex | $P_1^+ \times P_1$ | 2/1 | 2/1.5 | 2/1 | 2/0.5 | 2/0.5 | 1/0.5 | 2/0.5 |
| elements | $P_2 \times P_1$ | 3/2 | 3/2 | 3/2 | 3/0.5 | 3/0.5 | 1/0.5 | 2/0.5 |
| | $P_2 \times P_0$ | 2/1 | 2/1 | 2/1 | 2/1 | 2/1 | 1/0.5 | 2/1 |
| | $P_2^+ \times P_{-1}$ | 3/2 | 3/2 | 3/2 | 3/2 | 3/2 | 1/0.5 | 2/1.5 |

## 5 Conclusions

Historically as well as recently, geodynamics codes that solve the Stokes equations have based their numerical methods on a wide variety of finite element discretisations – nearly every element ever invented has been used in some geodynamics code or other. This diversity of approaches may not always have been motivated by careful considerations of what the best method is, but also human elements such as what the implementer was familiar with or felt feasible to implement. At the same time, today's finite element discretisation libraries upon which most new codes are built support a broad range of elements, both low and high order, and as a consequence, evidence-based decisions about which element to use are now both possible and called for. As a consequence, comparative studies such as the current one for simplex elements and the first part of our work in Thieulot and Bangerth (2022) for hypercube elements are both useful and necessary.

Having compared a number of possible finite element choices for the Stokes equations using a carefully selected set of benchmarks, we can summarize our findings as follows:

- **The $P_1^+ \times P_1$ element is not accurate enough:** Although appealing on paper because of its stability and small number of unknowns, the $P_1^+ \times P_1$ element is also the least accurate one in most benchmarks.

- **The $P_2 \times P_0$ element can not accurately represent the lithostatic pressure:** Similarly, the $P_2 \times P_0$ element appeals because of its small pressure space and the fact that it is mass conservative due to the discontinuous pressure. At the same time, the low-order pressure does not allow the velocity to reach the optimal convergence rate, and using a piecewise

constant pressure simply does not result in sufficient accuracy for applications in which an accurate representation of the lithostatic pressure field is important – say, for problems with pressure-dependent rheologies.

– **Only Taylor-Hood elements are accurate and robust:** As a consequence of these considerations, only the Taylor-Hood-type elements $P_2 \times P_1$ and $P_2^+ \times P_{-1}$ are truly competitive across all applications we have considered. This is in line with the conclusions of the first part of this work (Thieulot and Bangerth, 2022) where we have found that on hypercube cells, only the $Q_2 \times Q_1$ and $Q_2 \times P_{-1}$ elements are consistently able to provide sufficient accuracy across benchmarks. This is despite the non-trivial costs of these elements due to their large number of velocity degrees of

freedom, in particular in 3d, and consequent large number of nonzero entries in system matrices – apparently this is a price one needs to pay for consistently high accuracy.

    – **It is not obvious which of the Taylor-Hood variants is better:** Comparing between the two Taylor-Hood-like elements on triangles, the $P_2 \times P_1$ element provides a substantially better pressure approximation than the $P_2^+ \times P_{-1}$ element for smooth solutions; in other cases, the difference is marginal, and in yet other cases the discontinuous pressure elements

are substantially better. In essence, the difference is not universally large enough either way to recommend one over the other based on accuracy alone. The same is true when considering run times: All of the Taylor-Hood-type elements take about the same time in the assembly of the matrix, and in the solution of the linear system; run time is consequently not a criterion to choose one Taylor-Hood variant over another. On the other hand, if local mass conservation is important, or if one wanted to use a linear solver that can exploit the block diagonal structure of the pressure mass matrix of the

$P_2^+ \times P_{-1}$ combination, then this element may have a benefit over the $P_2 \times P_1$ element.

    – **Per degree of freedom, hypercube elements are slightly more accurate than the corresponding simplex elements:** Comparing between the Taylor-Hood-type elements on simplex and hypercube meshes, the $P_2 \times P_1$ element is typically less accurate than its counterpart $Q_2 \times Q_1$. Likewise, the $P_2^+ \times P_{-1}$ element is typically less accurate than its counterpart $Q_2 \times P_{-1}$ for smooth solutions. In neither case are the differences very large, however.

These conclusions conform with the results of the first part of this study: At the end of the day, only Taylor-Hood-type elements are consistently able to provide reliable and robust accuracy in geodynamics applications, not because they are inherently superior, but because all of the other choices fail on one benchmark or other in a way that make them unsuitable for the task. It is reassuring that this conclusion is the same for simplex and hypercube elements as this hints at the universality of the properties of finite element families, regardless of the choice of reference cell.

The comparisons we have made also support another conclusion: While triangular and tetrahedral meshes have rightfully been dominant in engineering applications for their ability to mesh complex geometries (and perhaps situations in which coefficients jump at predictable locations), they are generally slightly less accurate than the corresponding finite element on quadrilateral and hexahedral cells. Taking into account that they typically lead to matrices with fewer entries, one can speculate that per unit computational cost, their performance in terms of error as a function of computational work is roughly comparable

to that of hypercube cells. But, given that geodynamics applications oftentimes do not need complex geometries, this also

implies that simplex meshes and elements offer no specific benefit over hypercubes, and that there is no reason to abandon the common practice in the field to build codes based on hypercube cells.

*Author contributions.* CT conceived the study and ran all models. WB provided the theoretical finite element background, as well as the framing of the study. Both authors discussed the results and and jointly wrote the manuscript.

*Competing interests.* The authors declare that they have no conflict of interest.

*Acknowledgements.* W.B. gratefully acknowledges support by the National Science Foundation through awards OAC-1835673 as part of the Cyberinfrastructure for Sustained Scientific Innovation (CSSI) program, and EAR-1925595. We also thank the Computational Infrastructure for Geodynamics (CIG) for their long-term support of the ASPECT code that was used in the first part of this study and that has led us to the line of questions we have been investigating here.

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
