# Peer review of "On the choice of finite element for applications in geodynamics. Part II: A comparison of simplex and hypercube elements"

_EGUsphere, 2024_

## Author Comment (AC1)

**Reply to reviewers**

Below, please find a reply to the points raised by the reviewers. Issues not specifically discussed below have simply been fixed in the revision as suggested by the reviewers.

We would like to express our gratitude to the reviewers for the careful reading of our manuscript, along with the thoughtful comments. Peer review makes papers better, and this observation applies to the current one as well.

**Reviewer 1**

This manuscript is a useful contribution for the developers of the finite element codes for geodynamic applications. Despite various pieces of a stable discretization puzzle were circulating in the community during decades, this work (together with its first part considering quadrilateral/hexahedral elements) represents the only attempt to bring them together in an integrated picture. Potential benefits for the community include increased general awareness about selection of an efficient/robust discretization, which can in turn lead to substantial time savings in geodynamic software development.

Thank you for the supportive words!

In general, I would certainly recommend publishing this paper after the following issues/suggestions are addressed/incorporated.

[1] Using an extensive set of benchmarks, the authors demonstrate that only sufficiently rich interpolation functions (e.g. at least quadratic for the velocity) are able to perform well in a geodynamically relevant context on the simplex geometries. This result is quite expected. It is surprising, however, that little difference was found between the elements with continuous and discontinuous pressure (according to the conclusion section). This clearly contradicts previous findings of Pelletier et al. (1989), who have explicitly advocated superiority of discontinuous pressure approximations. I believe it is at least worth citing this paper in the context of this study, and discussing the difference in the conclusions.

First, thank you for bringing this paper to our attention. It is an interesting historical artifact illustrating that many of the ideas we today take for granted (such as scaling the different blocks of the Stokes matrix) were still new in the 1980s.

Pelletier et al. indeed recommend the use of discontinuous pressure elements, even for the relatively straightforward isoviscous cases they consider. Yet, this recommendation does not seem to have carried through to today's Computational Fluid Dynamics community which happily uses continuous elements. We are then left with trying to square these historical observations over the course of the intervening 37 years since the paper was submitted. In other words, what has changed, and how is the paper relevant to us today?

Our best interpretation is that the meshes we use have become much finer. Pelletier et al. do not state what meshes they use, but one might guess from historical context and the figures

they show that their meshes used perhaps 20x10 cells, i.e. *h=0.1*, or at least somewhere there-about. In contrast, our experiments start with 16x16 meshes. In other words, if we want to understand Pelletier et al.'s results, we have to look at the approximation errors *at the very right edge* of our figures. In this region, the differences between all methods become quite small – which perhaps also explains why the development of non-conforming and low-order methods was more popular in the 1980s than it is today.

A separate interpretation is that Pelletier et al.'s recommendations stem from the fact that they were mainly interested in the *divergence* error because of its manifestation in solutions that were manifestly not physical. The divergence error converges to zero at the same rate as the error in the pressure (and the error in the gradient in the velocity), so it does not *vanish* compared to the error in the pressure as the mesh is refined. In practice, however, it appears to be small enough that practitioners no longer seem to worry about it as much as they used to (whether that is justified or not). In the context of geodynamics computations, we are today concerned about pressure errors because we are considering material models in which the rheology strongly depends on the pressure (at least close to the surface) and so need it to be accurate. The divergence error affects time dependent problems because of how it influences quantities that are advected along; in practice, however, it does not seem to be a large problem and practitioners seem to be happy with the level of error in the divergence, in contrast to the error in pressure and velocity/strain/stress.

In the end, we think that the conclusions of Pelletier et al. are not easy to interpret from today's perspective. We now mention the paper in the second-to-last paragraph of the introduction, but decided to limit how deeply we want to engage in a historically correct interpretation of their results.

The conclusion "there is little difference between the Taylor-Hood variants" was most like made based on the results of SolVi benchmark alone. I think this is slightly biased, since all elements performed equally bad in this benchmark. On the other hand, in the SolCx benchmark the discontinuous pressure elements performed much better compared to the continuous counterparts, even if viscosity jumps were not aligned with the element edges. I think this also deserves to be mentioned in the conclusions.

For the benchmarks in the current study, we see the following results:

- Donea-Huerta: The element with a continuous pressure performs slightly better.
- SolCX: The element with a discontinuous pressure performs substantially better (on structured and unstructured meshes).
- SolVi: On structured meshes, the element with continuous pressure is substantially better; on unstructured meshes, the element with discontinuous pressure is substantially better.

The point we wanted to make in the Conclusions is simply that it is at least not *obvious* that the discontinuous pressure is superior. This also matches a corresponding conclusion we had drawn for hypercube elements in the first part of the study.

We have reworded the beginning of the paragraph in question to make it clear that it is not obvious which element is better; the remainder of the paragraph already states that there are other criteria one can use to break the tie.

Altogether, I actually believe that both these benchmarks are not really capable of demonstrating the difference between the continuous/discontinuous pressure elements. I suggest to complement this study with a Rayleigh-Taylor instability benchmark with a high density and high viscosity layer overlaying a low density and low viscosity layer. The viscosity contrast should be large enough (e.g. comparable with the one used in SolCx benchmark).

We have carried out the proposed benchmark for viscosity contrasts ranging from 1e-2 to 1e+2 and found that all elements performed well with the exception of the $P_2 \times P_0$ element pair. Our results can now be found in the new Section 4.5, and largely match those of Section 4.4 (the "sinking block" benchmark) where we have already found that elements with constant pressures do not perform well on buoyancy-driven setups. This is consistent with what we found in the previous paper.

That said, the benchmark does not help much in resolving the question whether continuous or discontinuous pressures are better in Taylor-Hood-type elements. For the quadrilateral elements, a continuous pressure does better on very coarse meshes; for triangular elements, a discontinuous pressure appears slightly better. The benchmark is not well suited, however, to resolve the question at appropriately refined mesh sizes.

I also recommend authors not to term a conforming Crouzeix-Raviart element as a Taylor-Hood variant, especially in the conclusions section. Historically Taylor-Hood elements explicitly implied a continuous pressure interpolation in the geodynamic community. Significant confusion may result from the sentences: "Only Taylor-Hood elements are accurate and robust" and "There is little difference between the Taylor-Hood variants". Please rephrase them.

It is probably debatable what people consider a "Taylor-Hood" element or not. In substantial parts of the Computational Science and Engineering community, the term "Taylor-Hood" is generally understood to be a Stokes element with one degree higher for the velocity than the pressure, regardless of whether the pressure is continuous or not. Many books on the finite element method use the term in this way, for example. This is the way we have used the term, and have documented this choice in footnote 3; see also footnote 6 for what the term "Crouzeix-Raviart" might mean.

In short: It is no longer possible to use either Taylor-Hood or Crouzeix-Raviart in a historically correct way. Neither of the two papers in question propose unambiguously what we generally understand these terms to mean today, nor is there universal agreement among practitioners what a specific term might imply; the best anyone can do is clearly define within the paper what we mean by a term. We do that in these footnotes.

[2] To complete the overview presented in this study I believe it is necessary to include a discourse on the use of the so-called isoparametric elements. In geodynamic codes it is quite common to interpolate the velocities and coordinates with the same shape functions (e.g. Dabrowski et al., 2008). Stable elements which include mid-side and mid-face nodes, and use at least second order shape functions, can potentially allow elements with curvilinear edges and faces. This configuration can be achieved either when this geometrical flexibility is used on

purpose to fit curvilinear boundaries, or in the context of an Arbitrary Lagrangian Eulerian (ALE) advection scheme, when nodal coordinates are updated using the calculated velocities.

Yes, this is correct. Though one doesn't need to restrict oneself to *iso*parametric mappings: For example, the default mapping in the ASPECT code we used for the first part of our study uses Q4 (quartic) mappings, regardless of the elements used – in other words, it uses *supra*parametric elements.

We purposefully did not include a discussion of mappings because that would have made the number of possible choices even larger. We also think that it does not actually add anything to whatever conclusions one can draw from what we have done: If an element is not competitive on straight-sided triangles, there really is no reason to believe that it would be competitive if only one allowed curved edges. That is certainly so for the domains we consider herein where the domain was naturally bounded by straight lines and the only reason to include curved edges may have been for cases where there are internal interfaces – but even in cases where there are no such internal interfaces, the elements we eliminate as not competitive fare poorly.

In other words, we think of our paper as a filter that eliminates some elements based on poor performance. It is conceivable that a study of using curved edges will reveal that one or the other among the elements we find competitive (namely, the Taylor-Hood variants) is better. But we would like to leave this for a later study.

We have added a discussion of this matter to the end of Section 3.3, and also renamed that section to make it clear which topics are not discussed in this paper.

At least for the mixed rectangular elements with discontinuous pressure the inf-sup stability estimates are not available for the curvilinear shapes, and special measures need to be taken to restore the convergence (e.g. Chilton, and Suri, 2000). The numerical experiments with more traditional displacements-based elements also suggest that curved edges should be avoided (e.g. Lee and Bathe, 1993). To my knowledge, there is currently a lack of similar studies in the available literature with a focus placed on simplicial element shapes (triangles and tetrahedra). It is also clear that this work is not affected by this issue, since only the elements with straight edges are considered.

Yes, indeed – it is not widely known that the inf-sup estimates do not easily generalize to curved elements. We consider this another reason to leave investigating such elements to a separate study.

As for the paper by Lee and Bathe: We read its results differently. Among its conclusions, the first is that serendipity-type elements are inferior to Lagrange elements – a conclusion that at the time (more than 30 years ago) may have been disappointing because memory and CPU time was so much more of a concern at the time than accuracy that serendipity elements were widely used because they had fewer degrees of freedom. But that is not particularly relevant any more today because these elements have largely fallen out of favor because of their overall poor accuracy. The paper does stress that using curved edges at the boundary is acceptable where necessary. It mentions that curved edges in the interior should be avoided, but perhaps that is a point that also has lost some of its relevance. This is because what ultimately matters is the degree of curvature edges experience, and that is related to the ratio of cell diameter and the diameter of the objects (say, of inclusions) one tries to mesh via curved edges. It is obvious

from the pictures in Lee and Bathe that at the time, this was a much more urgent problem: For example, the meshes in Figures 15 and 16 show a quarter circle meshed with 2 and 12 edges, respectively, each incurring substantial curvature; today, such studies would be done with 10-100 times higher resolution, resulting in cells that to the naked eye will have straight edges even if in the actual code they are curved. As a consequence, the effects of mesh distortions due to curved edges were much greater than today, and we think that the study's importance has perhaps diminished somewhat.

We acknowledge that the issue has not entirely gone away. For example, in other work, we are currently investigating large numerical errors that result from trying to mesh the entire Earth taking into account actual surface topography – where surface features to be resolved are smaller than the mesh size we are able to use and will likely be for a long time. In practice, we address this by smoothing the surface topography on length scales larger than the mesh size. The effects we see might very well be similar to the ones by Lee and Bathe. But we think that they are orthogonal to the ones we study herein and should be left for later investigations (see also below).

Nevertheless, I believe that stability of the curvilinear elements should be also covered in this study, since potential side effects might be relevant for the community. If possible, a test should be designed to demonstrate the influence of element curvature on the convergence rates. In any case the text should contain a discussion on this topic with potential remedies explicitly indicated. The latter can include, for example, the use of sub-parameric elements, e.g. when only the linear shape functions and corner nodes are used to represent the element geometry, and the element edges and faces remain straight/planar.

As mentioned, we would like to defer such a study for a separate paper. Ours is already at 29 or so pages, and we think that it serves as a good filter to eliminate some element choices from consideration. It might make for an interesting study to compare the various Taylor-Hood elements on both hypercube and simplex cells with curved edges, but this is clearly orthogonal to what we are doing here.

That said, as mentioned above, we have added a short discussion of the matter to the end of Section 3.3.

[3] This manuscript only presents the two-dimensional tests performed with triangular elements. I disagree with the argumentation given by the authors to exclude the three-dimensional tests and tetrahedral elements. The convergence results cannot be trivially extrapolated from 2D to 3D. It cannot be a priory guaranteed that the same type of discretization exhibits the same convergence rate, is equivalently stable, or even exists both in 2D to 3D (e.g. wedge elements). I believe this study would still profit from including tetrahedral elements.

Perhaps, but again we think this is a topic for a separate study. This is particularly true because doing the sort of computations we do here in 3d is so much more computationally expensive and likely out of range of what we can achieve with our single-threaded Python-based solver.

It is of course true that *a priori* it is not clear that our 2d results readily generalize to 3d. At the same time, we do not think that there is much evidence that they shouldn't: There is no common lore in the Computational Science and Engineering community that elements behave differently

in 3d than in 2d, and our combined four or so decades of experience also have not provided us with suspicions that they might. As a consequence, we really don't think that we would actually obtain anything different; at the same time, duplicating all results one more time for the 3d case would make the paper substantially longer.

References

Chilton, L., Suri, M., 2000. On the construction of stable curvilinear p version elements for mixed formulations of elasticity and Stokes flow. Numerische Mathematik, 86, 29-48.

Dabrowski, M., Krotkiewski, M., Schmid, D., 2008. MILAMIN: Matlab based finite element solver for large problems. Geochem. Geophys. Geosyst., 9, Q04 030

Lee, N.-S., Bathe, K.-J., 1993. Effects of element distortions on the performance of isoparametric elements. International Journal for Numerical Methods in Engineering, 36, 3553-3576.

Pelletier, D., Fortin A., Camarero, R., 1989. Are FEM solutions of incompressible flows really incompressible? (or how simple flows can cause headaches!). International Journal for Numerical Methods In Fluids, 9, 99-112.

Citation: https://doi.org/10.5194/egusphere-2024-1668-RC1

**Reviewer 2**

This article is complementary to recent work by the same authors that focused on different choices of hypercube (quadrilateral/hexahedral) finite elements for solving the Stokes equations in geodynamic modeling. This time, the authors focus on qualitatively assessing the accuracy of simplex (triangular/tetrahedral) finite elements. Interestingly, this work clearly shows that Taylor-Hood-type elements are the most accurate —and essentially the only viable option despite their higher computational cost— for solving Stokes equations with finite elements.

Given the lack of any previous exhaustive analysis of this type, I believe this is a relevant study that provides valuable insights to the computational geodynamic community. The paper is also well-written and easy to follow, and the results are clearly presented. Therefore, I recommend this paper for publication in Solid Earth. Please find below some minor comments:
We appreciate the kind words about our work!

- The authors discuss the performance of various elements. While it's relatively straightforward to infer which element should be faster, I believe the paper would benefit from a figure

comparing the performance of the assembly and linear solve for these elements at similar resolutions for one of the benchmarks. Metrics like DoFs/s could be useful for this comparison. Although the performance may not be entirely reliable since they are implemented in Python and likely lack many optimizations, they would still provide valuable qualitative insights into the relative speeds of the different elements.

We have added the requested run time data for the Donea-Huerta benchmark in Section 4.1; see the discussion and figures at the end of that section. The data shown there is that the cost of different elements is of course different, but not in a way so that the convergence rates shown as a function of $h$ would change when shown as a function of the run time. In particular, there appears to be very little difference between the various Taylor-Hood variants – and certainly not enough to elevate one over the others.

- It would be fantastic if you could upload the source files to a public repository. They would be highly educational and would greatly enhance reproducibility for anyone looking to compare future benchmarks against these results.

The code and its run scripts are now available at the following website: https://github.com/cedrict/fieldstone/tree/master/python_codes/fieldstone_120
If/after the paper is accepted, we will also upload the final version of our codes to Zenodo.

L29: "were the" should be "are the."
Fixed.

L215: I understand that expressing the resolution as h gives a better mental picture. However, since you are testing very different elements, perhaps using degrees of freedom instead of h is a more fair comparison?
Yes, we agree that this might have been a perhaps better way of showing the results. Yet, we would like to keep the figures as they were, for two reasons: (i) The first part of the paper uses $h$ as the $x$-axis on figures, and we would like to keep the two papers comparable; (ii) the mathematical literature (correctly) uses $h$ to derive convergence rates rather than the number of degrees of freedom and everyone is accustomed to the fact that a quadratic element has convergence rate $O(h^3)$ in the $L_2$ norm, rather than using $O(N^{-3/d})$ – note the dependency on the number of space dimensions to get the rate right.
Separate from this is the fact that neither $h$ nor $N$ provide good measures of "cost". A quadratic element in 2d has slightly more than twice as many degrees of freedom as a linear element, but the cost for assembly is not twice as high – it is about five times as much because one has to compute a dense 9x9 matrix rather than a 4x4 matrix in the scalar case. The same applies for the direct solver we employ: Its cost likely grows as $O(N^{3/2})$ with the overall number of unknowns, not even taking into account that the matrices for quadratic elements are substantially denser than the ones for linear elements, or that the mass matrix for discontinuous elements is block diagonal whereas it is not for continuous elements. In practice, we find growth rates that are even slightly higher than the 3/2 mentioned above – see Figures 4 and 5 in the current manuscript, for example.

In the end, that means that it is quite difficult to define a "fair" measure. Moreover, what we find in this paper is that there are certain *categorical* differences between the elements in that they have different convergence orders. These are reflected in the graphs whether one shows results as a function of *h* or as a function of *N* (or $N^{-3/2}$) – corresponding graphs would simply be translated up or down (or left/right, if you prefer). The *position* of the lines would change, but the categorical differences would remain as shown.

That all said, we added (an abridged version of) this kind of discussion to the paragraph in question in Section 3.4. Furthermore, we also added a comparison of elements in terms of CPU time (perhaps the only useful measure besides *h*) as discussed elsewhere in this reply to reviewers; this addition can be found at the end of Section 4.1 (the Donea-Huerta benchmark).

L215: "h is a same" should be "h is the same."
Fixed.

L225: "python" should be "Python."
Fixed.

L278: I assume the inclusion viscosity is larger than that of the matrix; could you slightly rephrase this to make it fully clear to the reader? The analytical solution of this problem also allows for different inclusion radii; could you add to the text what value you used? Additionally, could you specify the size of the domain to provide a better idea of the equivalence between the number of elements and h?
Indeed. We have adjusted the text to the following:
"It [the SolVi benchmark] models a situation where the viscosity inside an inclusion is 1000 times larger than outside the inclusion, and where we make no attempt at resolving this boundary with the structured mesh -- similar to realistic situations of slab subduction or other cases of large and perhaps dynamically changing viscosity jumps that cannot practically be resolved using the meshes in use. The setup is identical to the one in Thieulot and Bangerth (2022) although here we only model one quadrant of the problem: The domain is a unit square $(0,1)^2$, the inclusion is centered on the origin, and the analytical velocity is prescribed on all sides."

L279: "the (structured) mesh" should be "the structured mesh."
Fixed.

L295: Isn't P2 × P-1 performing better than P2 × P1 for the unstructured mesh?
Yes, of course. We have added this statement: "A comparison of the curves for the structured meshes shows that, for this complex situation, the Taylor-Hood elements $Q_2 x Q_1$ and $P_2 x P_1$ fare the best, at least as far as "error for a given mesh size" is concerned. For the very specific discontinuity-aligned unstructured meshes, the $P_2^+ x P_{-1}$ pair emerges as the overall best with a quadratic pressure error convergence."

Figure 6: P2 × P0 (red dots) seems to have some duplicated data points.

Sort of 🙂 For each viscosity ratio, we obtained measurements using three density ratios $\delta\rho/\rho_{fluid}$. From a theoretical perspective, these three data points should superimpose on top of each other. Indeed, for most elements they do – but for the $P_2 \times P_0$ they do not and so it looks like we have duplicated points. We added a sentence to clarify this at the end of the introduction of Section 4.4 and refer to it from the caption of Fig. 6.

This is perhaps a topic for a conversation over a glass of beer. The actual process of refining a quad/oct-tree-based mesh is not overly difficult and will incur about the same level of programming complexity as refining a simplex mesh. In fact, keeping the quad/oct-tree is not difficult either: In its simplest form, each cell simply needs to store a pointer to its parent cell as well as pointers from parents to children.
But the comment is probably directed towards *adaptive* or *local* mesh refinement. There is, undoubtedly, some complexity in using hanging nodes. In deal.II, for example, the generic implementation of hanging nodes takes about 3000 lines of code, plus a couple of dozen lines in each finite element class to compute the element-specific interpolation matrices from child to parent face. This is to be compared to a total of nearly a million lines of code in the entire library. It is also the result of optimizing the implementation over more than 20 years – we think that one could probably implement the algorithm in under 1000 lines, but it would not run in parallel, not support hp-adaptive refinement, and not be as efficient with CPU time and memory.
At the same time, there is also very substantial complexity to using local refinement on simplex meshes. Specifically, one either needs to use longest-edge or red-green refinement strategies to avoid degenerating meshes. Neither of these is at all trivial to implement for three-dimensional meshes whereas the logic for hanging nodes is the same between 2d and 3d.
In the end, our experience is that implementing hanging nodes is not *that* complex and that perhaps this should not be a criterion to choose between the two types of meshes.

Thank you, the DOI we had used was a temporary one that was later replaced when the page number (i.e. 229) was known.

**Unsolicited review by Marcus Mohr**

Dear authors,

I just finished reading your preprint "On the choice of finite element for applications in geodynamics. Part II: A comparison of simplex and hypercube elements". Like Part I, I found

this really interesting and very much appreciate that you extended your testing to simplex elements.

First: We really appreciate this review. It is not common that someone who happens onto a paper under review takes the time to write a review for it, and then sends it to the authors. We genuinely appreciate the feedback – thank you!

 In addition to the comments by the two reviewers, I'd like to ask some simple questions and make some small comments.

- Your conjecture on the performance on non-conforming elements would be supported (at least partially) by Terrel et al. (2012) [a chapter in "The FEniCS Book"]. There different simplicial element pairs are evaluated for a smooth 2D Stokes problem. A lowest order non-conforming Crouzeix-Raviart element for velocity with a P0 pressure is also included. It does not perform well, basically similarly bad for velocity than the mini-element and even worse for pressure. Excellent for divergence, though, as to be expected.

Thank you for the reference, we have added references to it in the introduction (bottom of p. 2).

- On line 208 you state that "None of the computational experiments we perform herein presents the problem of a velocity nullspace". However, the SolCx benchmark in Part I is described as having "free-slip on all edges", same for the Sinking Block (line 313). Would this not result in a non-trivial kernel for the velocity block in the matrix and, hence, a non-trivial nullspace?

No, this is not right – at least not for the velocity. When prescribing "free-slip", one states that the normal velocity is zero and the tangential traction is zero. Any external pressure one might want to include in traction boundary conditions only contributes in the normal direction, of course, and similarly the boundary term that results from integration by parts is only the tangential component of the stress at the boundary and so does not include the pressure. In other words, adding or subtracting a constant to the pressure is within the kernel of the weak formulation. That is, there is a non-trivial *pressure* nullspace.
But there is no *velocity* nullspace. The left and right boundaries allow for the addition of a vertical velocity, but this is clearly no longer possible when taking into account that we require the velocity to be horizontal at the top and bottom boundaries. In other words, none of the rigid body motions are in the kernel of the boundary conditions.

- In the SolVi case, I assume you have just used the actual viscosity at the quadrature points? Because there was no real superior case in the different averaging settings you compared in Part I, if I remember that correctly?

Yes, the viscosity and density/buoyancy terms are evaluated at each quadrature point.

- The remark on line 129 that the bubble is a "quadratic polynomial" to me seemed a little misleading? As a product of the natural coordinates should the degree of the bubble not be (dim+1)?

Indeed. We have corrected this in the text.

- Line 313 says the domain is the unit "cube". Should that be unit "square" with the latter indicating (-1,1)x(-1,1) as otherwise part of the sinker would be outside the domain, if it was (0,1)x(0,1)?

Of course, thank you.

- The DOI for the 1973 Crouzeix-Raviart paper would be https://doi.org/10.1051/m2an/197307R300331 (I looked that up. Actually the footnotes on the naming/history for TH and CR were very nice :)

Yes, it's a nice example that the names we attach to things are often not historically accurate.

- "Tayloor" on p.5 should be "Taylor" same in the references.

Oh, indeed. Thank you for pointing it out!

- W.r.t. the remark (line 150) on the P2 x P0 element being "the cheapest stable element with discontinuous pressure" my colleague Fabian Böhm reminded me that there exists the EG-P0 pair, where a P1 element for velocity is enriched by a single discontinuous vector-valued component. Hence, one has only one additional degree of freedom. See e.g.
  - Si et al. (2022), SINUM, https://doi.org/10.1137/21M1391353
  - Si et al. (2022), arXiv, https://doi.org/10.48550/arXiv.2110.05310
  Fabians master thesis contains some tests for this pairing, e.g. a comparison of EG-P0 and P2-P1 for SolVi:
  https://www10.cs.fau.de/publications/theses/2023/Master_B%C3%B6hmFabian.pdf

That is an interesting find, and fun to read about! Thank you for the references.
The EG-P0 element uses an enriched linear space for the velocity in which the six linear velocity shape functions (three for each vector component) are augmented by a single discontinuous shape function. If one takes a uniform triangular mesh with cells of optimal shape, then each vertex is shared by six cells. Asymptotically, each cell then "owns" one sixth of the continuous shape functions, plus the discontinuous one – so two degrees of freedom per cell. In contrast,

for the $P_2$ space, we have 6 vertex degrees of freedom shared by 6 adjacent cells each, plus 6 face degrees of freedom shared by two adjacent cells each, resulting in each cell asymptotically "owning" three degrees of freedom: One more than the EG one. Obviously, as the comparison in the thesis of Böhm shows, the element is not competitive (perhaps not unexpectedly).

Of course, the EG element is discontinuous and consequently non-conforming. We think that the $P_2$x$P_0$ element is the cheapest *conforming* element, but it is not necessary to belabor the point. We have reworded the text accordingly.

Looking forward to the final version of the paper.